# On the stability of canonical correlation analysis and partial least squares with application to brain-behavior associations
Markus Helmer [1,2], Shaun Warrington [3], Ali-Reza Mohammadi-Nejad[3,4], Jie Lisa Ji[1,2,5], Amber Howell[1,5], Benjamin Rosand[6], Alan Anticevic[1,2,5,7], Stamatios N. Sotiropoulos [3,4] ✉ & John D. Murray [1,2,6,8] ✉

Associations between datasets can be discovered through multivariate methods like Canonical Correlation Analysis (CCA) or Partial Least Squares (PLS). A requisite property for interpretability and generalizability of CCA/PLS associations is stability of their feature patterns. However, stability of CCA/PLS in high-dimensional datasets is questionable, as found in empirical characterizations. To study these issues systematically, we developed a generative modeling framework to simulate synthetic datasets. We found that when sample size is relatively small, but comparable to typical studies, CCA/PLS associations are highly unstable and inaccurate; both in their magnitude and importantly in the feature pattern underlying the association. We confirmed these trends across two neuroimaging modalities and in independent datasets with $n \approx 1000$ and $n = 20,000$, and found that only the latter comprised sufficient observations for stable mappings between imaging-derived and behavioral features. We further developed a power calculator to provide sample sizes required for stability and reliability of multivariate analyses. Collectively, we characterize how to limit detrimental effects of overfitting on CCA/PLS stability, and provide recommendations for future studies.

Discovery of associations between high-dimensional datasets is a topic of growing importance across scientific disciplines. For instance, large initiatives in human neuroimaging collect, across thousands of subjects, rich multivariate brain imaging measures paired with psychometric and demographic measures[1,2]. A major goal is to determine the existence of an association linking individual variation in behavioral features to variation in brain imaging features and to characterize the dominant latent patterns of features that underlie this association[3,4]. One widely employed statistical approach to map multivariate associations is to define linearly weighted composites of features in both datasets (e.g., brain imaging and psychometric) with the sets of weights—which correspond to axes of variation—selected to maximize between-dataset association strength (Fig. 1). The resulting profiles of weights for each dataset can be examined for how the features form the association. Depending on whether association strength is measured by correlation or covariance, the method is called *canonical correlation analysis* (CCA)[5] or *partial least squares* (PLS)[6–11], respectively.

CCA and PLS are commonly employed across scientific fields, including genomics[12] and neuroimaging[3,4,13,14].

Analysis of such high-dimensional datasets is challenging due to inherent measurement noise and the often small sample sizes in comparison to the dimensionality of the data. Although the utility of CCA/PLS is well established, open challenges exist regarding stability in characteristic regimes of dataset properties. Stability implies that elements of CCA/PLS solutions, such as association strength and weight profiles, are reliably estimated across independent collections of observations from the same population. Instability or overfitting can occur if an insufficient sample size is available to properly constrain the model. Manifestations of instability and overfitting in CCA/PLS include inflated association strengths[15–19], out-of-sample association strengths markedly lower than in-sample[18,20], and feature profiles/patterns that vary substantially across studies[15,18–25]. Also, while some theoretical results for the sampling properties of CCA are available under normality assumptions[26], one generally needs to resort to resampling

[1]Department of Psychiatry, Yale School of of Medicine, New Haven, CT 06511, USA. [2]Manifest Technologies, New Haven, CT 06510, USA. [3]Sir Peter Mansfield Imaging Centre, Mental Health and Clinical Neurosciences, School of Medicine, University of Nottingham, Nottingham NG7 2UH, United Kingdom. [4]National Institute for Health Research (NIHR) Nottingham Biomedical Research Ctr, Queens Medical Ctr, Nottingham, United Kingdom. [5]Interdepartmental Neuroscience Program, Yale University School of Medicine, New Haven, CT 06511, USA. [6]Department of Physics, Yale University, New Haven, CT 06511, USA. [7]Department of Psychology, Yale University, New Haven, CT 06511, USA. [8]Department of Psychological and Brain Sciences, Dartmouth College, Hanover, NH 03755, USA. ✉e-mail: stamatios.sotiropoulos@nottingham.ac.uk; john.d.murray@dartmouth.edu

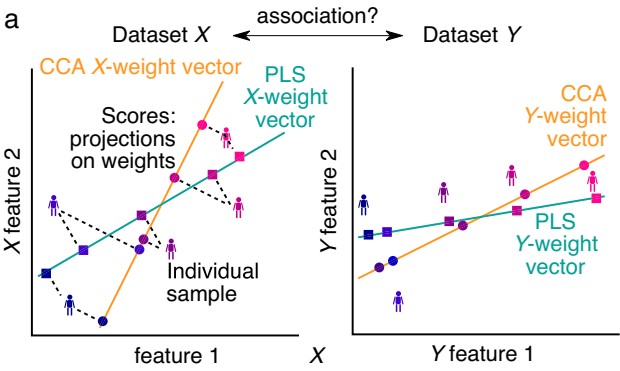

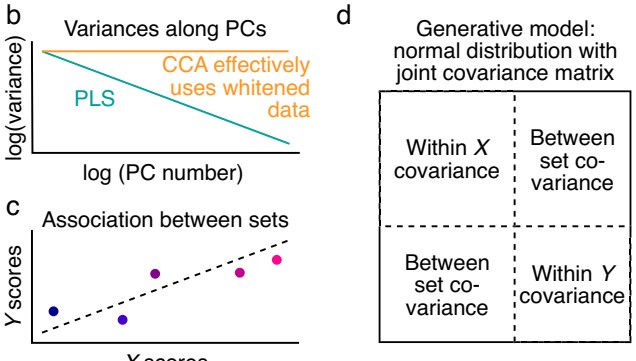

**Fig. 1 | Overview of CCA, PLS and the generative model used to investigate their properties. a** Two multivariate datasets, *X* and *Y*, are projected separately onto respective weight vectors, resulting in univariate scores for each dataset. The weight vectors are chosen such that the correlation (for CCA) or covariance (for PLS) between *X* and *Y* scores is maximized. **b** In the principal component coordinate system, the variance structure within each dataset can be summarized by its principal component spectrum. For simplicity, we assume that these spectra can be modeled as power-laws. CCA, uncovering between-set correlations, disregards the variance structure and can be seen as effectively using whitened data (cf. Methods). **c** The association between sets is encoded in the association strength of *X* and *Y* scores. **d** Datasets *X* and *Y* are jointly modeled as a multivariate normal distribution. The within-set variance structure (**b**) corresponds to the blocks on the diagonal, and the associations between datasets (**c**) are encoded in the off-diagonal blocks.

approaches to calculate uncertainty estimates like confidence intervals. Stability of models is essential for replicability, generalizability, and interpretability[27]. Therefore, it is important to understand how stability of CCA/PLS solutions depends on dataset properties.

In neuroimaging, it has recently been suggested that thousands of subjects are required to achieve reproducible results when performing multivariate "brain-wide association studies" as effect sizes are typically small[28]. This claim generated recent debate in the field[29–32]. A number of papers argue that larger effect sizes can be expected[29,30], that sample-size requirements can be reduced via focused designs and cohorts[31], and that cross-validation avoids inflated associations[32]. Yet, all previous studies and comments are mostly based on empirical findings and focus primarily on effect sizes. In the context of CCA/PLS, it remains unclear how elements of solutions differentially depend on dataset properties, and how CCA vs. PLS as distinct methods exhibit differential robustness across dataset regimes.

To investigate these issues systematically and go beyond empirical findings, we developed a generative statistical model to simulate synthetic datasets with known latent axes of association. Sampling from the generative model allows quantification of deviations between estimated and true CCA/

PLS solutions. We found that stability of CCA/PLS solutions requires more samples (per feature) than are commonly used in published neuroimaging studies. With too few individual observations, estimated association strengths were too high, and estimated weights could be unreliable and non-generalizable for interpretation. CCA and PLS differed in their dependencies and robustness, in part due to PLS weights exhibiting an increased similarity towards dominant principal component axes compared to CCA weights. We analyzed two large state-of-the-art neuroimaging-psychometric datasets, the Human Connectome Project[1] and the UK Biobank[2], which followed similar trends as our model. We also observed similar trends when considering features from two neuroimaging modalities, functional and diffusion MRI. These model and empirical findings, in conjunction with a meta-analysis of estimated stability in the brain-behavior CCA literature, suggest that discovered association patterns through typical CCA/PLS studies in neuroimaging are prone to instability. Finally, we applied the generative model to develop algorithms and a software package for calculation of estimation errors and required sample sizes for CCA/PLS. We end with practical recommendations for application and interpretation of CCA/PLS in future studies.

## Results

CCA/PLS describe statistical associations between multivariate datasets by analyzing their between-set covariance matrix (Fig. 1, Supplementary Note 1). A weighted combination of features called scores is formed for each of the two datasets, and the association strength between these score vectors is optimized by defining the weight vectors. CCA and PLS use Pearson correlation and covariance as their objective functions, respectively. (PLS is also referred to as PLS correlation [PLSC] or PLS-SVD[6–11].) We call the corresponding optimized value between-set correlation and between-set covariance, respectively. We also calculate loadings, which we define as the univariate Pearson correlations (across observations) between CCA/PLS scores and each original variable in the dataset. We note that alternative terminologies exist[4,8,13,33,34]. CCA/PLS scores (as described above) could also be called variates; weights (as described above) could also be called vectors; and loadings (as described above) could also be called parameters. For CCA, the correlation between the score vectors, i.e. the between-set correlations, are also called inter-set correlations or canonical correlations.

### A generative model for cross-dataset multivariate associations

To analyze dependencies of stability for CCA and PLS, we need to generate synthetic datasets of stochastic observations with known, controlled properties. We therefore developed a generative statistical modeling framework, GEMMR (Generative Modeling of Multivariate Relationships), which allows us to design and generate synthetic datasets, investigate the dependence of CCA/PLS solutions on dataset size and assumed covariances, estimate weight errors in CCAs reported in the literature, and calculate sample sizes required to bound estimation errors (see Methods).

To describe GEMMR, first note that data for CCA/PLS consist of two datasets, given as data matrices *X* and *Y*, each with multiple features and an equal number *n* of observations. We model the within-set covariance with power-law decay in the variance spectrum, which we constrain to empirically consistent ranges (Supplementary Fig. 1). GEMMR then embeds between-set associations by defining associated weight axes in each set. Finally, the joint covariance matrix for *X* and *Y* is composed using the within- and between-set covariances (Fig. 1d) and the normal distribution associated with this joint covariance matrix constitutes our generative model.

We systematically investigated the downstream effects on CCA/PLS stability of generative model parameters for dataset properties: number of features, assumed population (or true) value of between-set correlation, power-laws describing the within-set variances, and sample size. Weight vectors were chosen randomly and constrained such that the *X* and *Y* scores explain at least half as much variance as an average principal component in

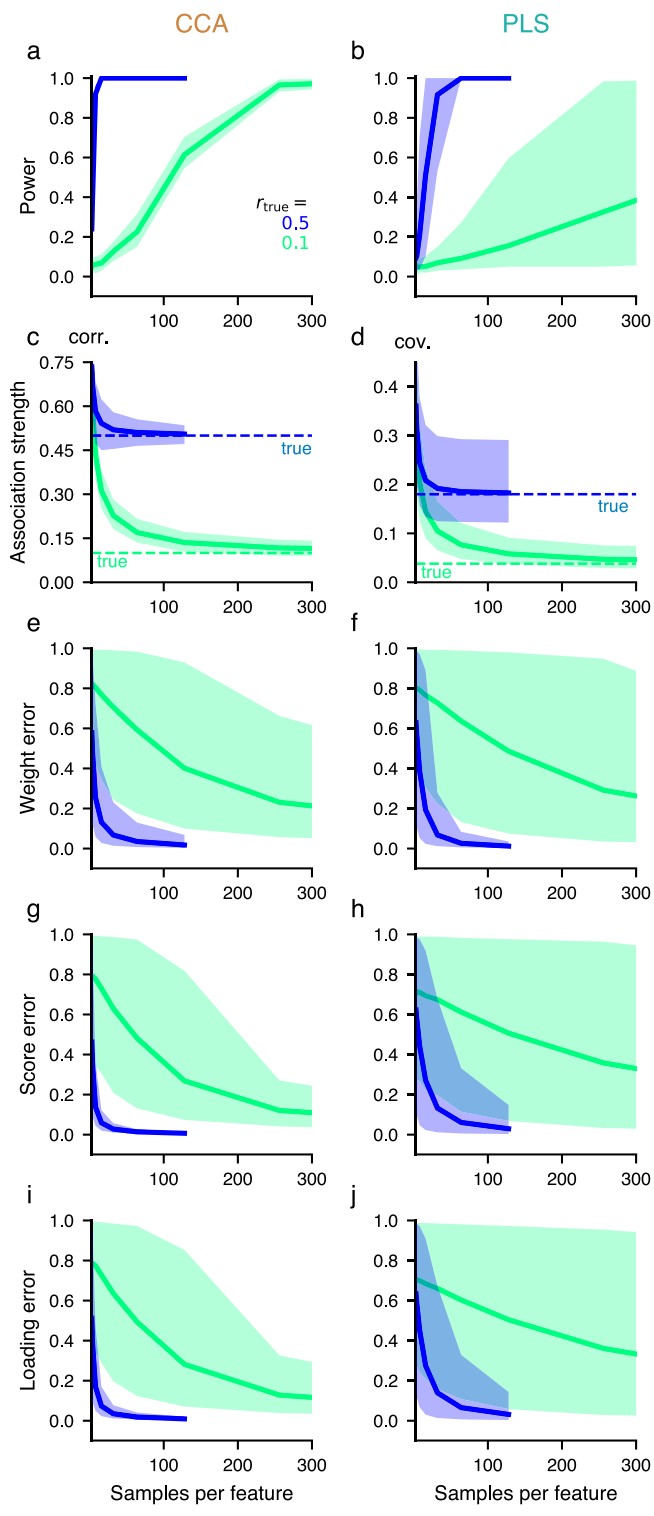

**Fig. 2 | Sample-size dependence of CCA and PLS. a, b** For sufficiently large sample sizes, statistical power to detect a non-zero between-set association strength converges to 1. Shaded areas show 95% confidence intervals across 25 covariance matrices representing distributions with the indicated $r_{true}$ but different (true) weight vectors. **c, d** In-sample (solid) estimates of the between-set correlations approach their assumed true (population) value (dashed). **e, f** Weight errors (quantified as the "1—absolute cosine similarity" between the true weights of the generative model and estimated weights from CCA/PLS on a collection of samples, separately for $X$ and $Y$ and taking the greater of the two), **g, h** score errors (measured as "1—absolute Pearson correlation" between estimated and true scores, which, in turn, are obtained by applying estimated and true weights to common test data) **i, j** as well as loading errors (measured as "1—absolute Pearson correlation" between estimated and true loadings) become close to 0 for sufficiently large sample sizes. Original data features are generally different from principal component scores, but as the relation between these two data representations cannot be constrained, we calculate all loadings here with respect to principal component scores. Moreover, to compare loadings across repeated datasets we calculate loadings for a common test set, as for CCA/PLS scores. Left and right columns show results for CCA and PLS, respectively. For all metrics, convergence depends on the true (population) between-set correlation $r_{true}$ and is slower if $r_{true}$ is low. Note that the color code indicates true (population) between-set correlation and corresponds to the dashed horizontal lines in c-d. Curves show mean and 95% confidence intervals of CCA/PLS estimates across 100 draws of collections of observations with a given sample size from 25 different generative models with the indicated $r_{true}$ but varying true (population) weight vectors (see Methods). $X$ and $Y$ feature space dimensionality was 8.

5 samples per feature (Supplementary Fig. 2a). A key parameter of the generative model is the population value, or true value, of the association strength, i.e., the value one would obtain, both through in-sample and out-of-sample estimation, as the sample size grows toward infinity. Importantly, like the mean of a normal distribution the population value of the association strength, $r_{true}$, is independent of the collection of samples and the sample size used to estimate it, but constitutes instead a parameter of the distribution from which observations are drawn. As such $r_{true}$ is a well-defined free parameter that can be varied independently of sample size.

We assessed, first, whether a significant association can robustly be detected, quantified by statistical power, and found relatively low power at typical sample sizes and population effect sizes (Fig. 2a, b). Second, we evaluated convergence of association strength (Fig. 2c, d). We calculated the (with-)in sample association strength by performing CCA/PLS with a given collection of samples, and out-of-sample association strength through cross-validation (see Methods). The observed between-set correlation converges to its assumed true (population) value for sufficiently large sample sizes (Fig. 2c, d). In-sample estimates of the association strength overestimate their true value (Fig. 2c, Supplementary Figs. 3 and 4). A sufficient sample size, depending on other covariance matrix properties, is needed to bound the error in the association strength. Cross-validated estimates underestimate the true value to a similar degree as in-sample estimates overestimate it[18] (Supplementary Fig. 5).

In addition to association strengths, CCA/PLS solutions provide weights that encode the nature of the association in each dataset, as well as scores which represent a latent value assigned to each individual observation (e.g. subject). Finally, some studies report loadings, i. e. the correlations between original data features and CCA/PLS scores (Supplementary Fig. 6a, b). We found that estimation errors for weights, scores, and loadings decrease monotonically with sample size and more quickly for stronger population effect sizes (Fig. 2e–j).

We used "samples per feature" as an effective sample size parameter to account for the fact that datasets in practice have very different dimensionalities. Others have previously explored the effect of varying samples and features[35,36]. Figure 3 and Supplementary Note 2 show that power and error metrics for CCA are parameterized well in terms of samples per feature, whereas for PLS it is only approximate. Nonetheless, as samples per feature is arguably most straightforward to interpret, we presented results in terms of samples per feature for both CCA and PLS[37].

their respective sets. For simplicity, we restrict our present analyses to a single between-set association mode. We use the term "number of features" to denote the total number across both $X$ and $Y$.

## Sample-size dependence of estimation error

Using surrogate datasets from our generative model, we characterized estimation error in multiple elements of CCA/PLS solutions. Here we use samples per feature as an effective sample-size measurement, which accounts for widely varying dimensionalities across empirical datasets. A typical sample size in the brain-behavior CCA/PLS literature is about

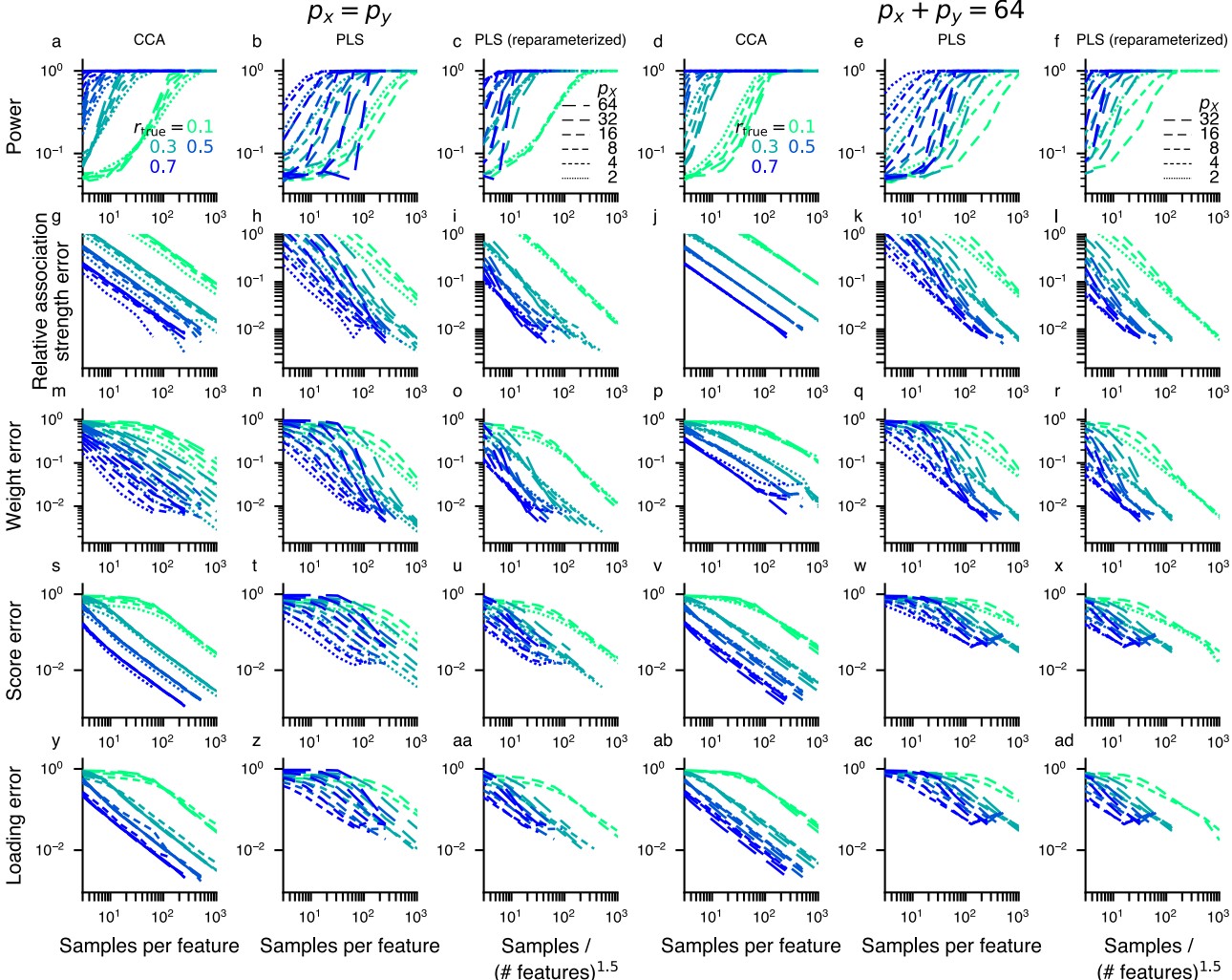

**Fig. 3 | Samples per feature is a key effective parameter.** The number of samples per feature is a key effective parameter. $p_X$ and $p_Y$ denote the number of features for $X$ and $Y$ respectively. Color hue represents true between-set correlation $r_{true}$, saturated colors are used for $p_X = 2$, and fainter colors for higher $p_X$ (in this figure, $p_X \in \{2, 4, 8, 16, 32, 64\}$), where $p_X$ ($p_Y$) refers to the number of features in the $X$ ($Y$) dataset. We fixed $p_X = p_Y$ in the left 3 columns, whereas we fixed $p_X + p_Y = 64$ (and thus had $p_X \neq p_Y$) in the right 3 columns. In CCA (first column), for a given $r_{true}$, power and error metric curves for various number of features are very similar when parameterized by samples per feature. In PLS (second column), the same tendency

can be observed, albeit the overlap between curves of the same hue (i. e. with same $r_{true}$ but different number of features) is worse. When "samples / (number of features)$^{1.5}$" is used instead (third column), the curves overlap more. The same trends can be seen in the right 3 columns, where $p_X \neq p_Y$. Curves in the first row are means across 25 covariance matrices representing distributions with the indicated $r_{true}$ but different weight vectors. Curves in all other rows are averaged across the same 25 covariance matrices and 100 draws of collections of observations of the sample size indicated on the x-axis. Panels **a–f**, **g–l**, **m–r**, **s–x** and **y–ad** show, respectively, power relative association strength error, weight error, score error and loading error.

## Weight error and stability

Figure 2 quantifies how sample size affects CCA/PLS summary statistics. We next focused on error and stability of the weights, due to their centrality in CCA/PLS analyses in describing which features carry between-set associations[3]. Figure 4a, b illustrates an example of how CCA/PLS weight vectors exhibit high error when typical sample-to-feature ratios are used. We systematically measured weight stability, i.e., the consistency of estimated weights across independent collection of samples, as a function of sample size. At small sample sizes, the average weight stability was close to 0 for CCA and eventually converged towards 1 (i. e. perfect similarity) with more observations (Fig. 4c). PLS exhibited striking differences from CCA: mean weight stability had a relatively high value with high variability across population models even at low sample sizes (Fig. 4d), where weight error is very high (Fig. 2f).

To assess the dependence of weight error on the assumed true between-set correlation and the number of features, we estimated the number of observations required to obtain <10% weight error (Supplementary Fig. 7). The required sample size is higher for increasing number of features, and

lower for increasing true between-set correlation. We also observe that, by this metric, required sample sizes can be much larger than typical sample sizes in CCA/PLS studies.

## Weight PC1 similarity in PLS

At low sample sizes, PLS weights exhibit, on average, high error (Fig. 2f) yet also relatively high stability (Fig. 4d). This suggests a systematic bias in PLS weights toward an axis different than toward the true latent axis of association (Fig. 4b). We quantified the PC similarity as the cosine similarity between estimated weight vectors and principal component axes. We found that, for a range of different number of features and true between-set correlations, weight similarity to PC1 was strong for PLS (but not CCA), with PLS weight vectors exhibiting strong bias toward PC1 especially for low sample sizes (Fig. 4e, f).

## Comparison of loadings and weights

In addition to weights, loadings provide a measure of importance for each considered variable[33,38]. We found that for CCA, stability and error of

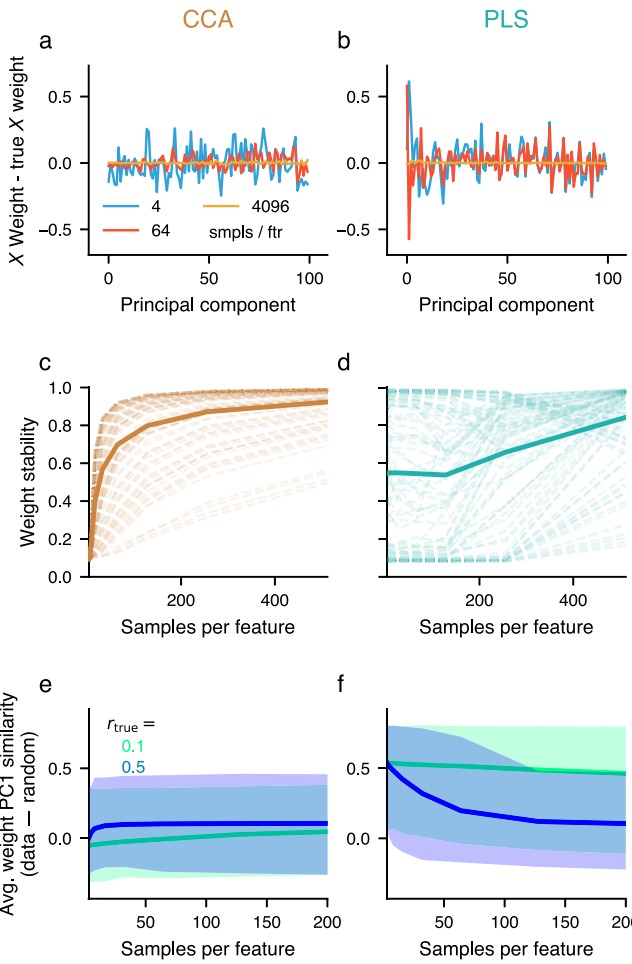

**Fig. 4 | Large number of observations required to obtain good weight estimates.**
**a**, **b** Realistic example where the true between-set correlation was set to $r_{true} = 0.3$. Estimated weights are close to the assumed true (population) weights, as long as the sample size is large enough. **b** For PLS even more observations were necessary. **c**, **d** Weight stability, i. e. the average cosine-similarity between weights across all pairs formed from 100 repetitions, increases towards 1 (identical weights) with more observations. For PLS, weight stability can be high, even with few observations. The true between-set correlation was set to $r_{true} = 0.3$. Each of the 100 dashed lines represents a different covariance matrix with different assumed weight vectors. The solid line shows the average across the dashed lines. **e**, **f** PC1 similarity was stronger for PLS (**f**) than for CCA (**e**) also for datasets with varying number of features and true between-set correlations $r_{true}$. Shown is relative PC1 similarity across synthetic datasets with varying number of features, relative to the expected PC1 similarity of a randomly chosen vector with dimension matched to each synthetic dataset. Shaded areas denote 95% confidence intervals across 6 feature space dimensionalities, 10 covariance matrices and 100 draws of collections of observations with the indicated sample size (*x*-axis) from the multivariate normal distribution associated with these covariance matrices.

loadings followed a similar sample size dependence as weights. In contrast, PLS loadings were extremely stable even at low sample sizes where error is high, indicating a strong bias (Supplementary Fig. 8a, b). For both CCA and PLS, a steeper power-law describing the within-set variance produced more stable loadings (Supplementary Fig. 8c, d). We next evaluated whether loadings exhibit bias toward principal component axes (Supplementary Fig. 8e, f). At small sample sizes, PLS loadings and weights, as well as CCA loadings, strongly resembled more dominant principal component axes. Thus, the within-set variance can have strong biases on CCA/PLS results, irrespective of true between-set associations (Supplementary Fig. 9).

## Empirical brain-behavior CCA/PLS

Do these phenomena from our generative modeling framework hold in empirical data? We focused on two state-of-the-art population neuroimaging datasets: Human Connectome Project (HCP)[1] and UK Biobank (UKB)[2]. Both provide multi-modal neuroimaging along with a wide range of behavioral and demographic measures, and both have been used for CCA-based brain-behavior mapping[2,3,39–43]. HCP data is widely used and of cutting-edge quality, and the UKB is one of the largest publicly available population-level neuroimaging datasets.

We analyzed two modalities from the HCP, resting-state functional MRI (fMRI) ($N = 948$) and diffusion MRI (dMRI) ($N = 1,020$, Supplementary Fig. 10a–d), and fMRI from the UKB ($N = 20,000$). Functional and structural connectivity features were extracted from fMRI and dMRI, respectively. After modality-specific preprocessing (see Methods), datasets were deconfounded and reduced to 100 principal components (Supplementary Fig. 11), following prior CCA studies[3,39–43]. We repeatedly formed two non-overlapping subsamples of subjects, varying sizes up to 50 % of the subjects, and assessed CCA/PLS solutions (Fig. 5, Supplementary Figs. 12 and 13).

In-sample association strength decreased with increasing size of the subsamples, but converged to cross-validated association strength clearly only for the UKB at large sample size (Fig. 5a, c, e, f). Figure 5a overlays reported CCA results from prior publications that used 100 features per set with HCP data, which further confirms the substantially decreasing association strengths as a function of sample size. HCP weight stabilities (Eq. 17) remained at low and intermediate values for CCA and PLS, respectively (Fig. 5b, d, f, h). In contrast, UKB weight stabilities reached values close to 1 (perfect stability). Moreover, for all datasets, PC1 similarity (Eq. 18) was close to 0 for CCA but markedly higher for PLS weights (Fig. 5b, d, f, h). Finally, loadings exhibited similar dependencies as weights, with higher PC1 similarity (Supplementary Fig. 14). Very similar behavior is observed when using very different features extracted from diffusion MRI (Supplementary Fig. 10a–d).

All these empirical results are in agreement with analyses of synthetic data discussed above (Figs. 2 and 4). The overall similarities between CCA/PLS analyses of different neuroimaging modalities and features (Fig. 5, Supplementary Fig. 10) suggest that sampling error is a major determinant in CCA/PLS solutions in typical data regimes. These results also show that stable CCA/PLS solutions with a large number of features can be obtained with UKB-size datasets.

We also explored reducing the data to different numbers of PCs than 100. Multiple methods have been proposed to determine an optimal number of PCs (see Discussion). Here, as an example, we used the max-min-detector from[44]. This method suggested 68 brain imaging and 32 behavioral dimensions for HCP[44], which yielded higher cross-validated association strengths and higher stabilities of weights. In UKB, we separately varied the number of retained neuroimaging and behavioral principal components and calculated CCA/PLS association strengths (Supplementary Fig. 15). We found that estimated association strengths rose strongly when retaining an increasing number of behavioral PCs, but only up to about 10. The situation for neuroimaging PCs differed between the methods, however. For CCA, retaining more neuroimaging PCs improved the association strength up to about 20–40 before plateauing. For PLS, on the other hand, the top PCs ($\approx 5$–10) were enough for the association strength to plateau. Altogether, these results demonstrate the potential benefits of careful, modality-specific dimensionality reduction strategies to enhance CCA/PLS stability.

## Samples per feature alone predicts published CCA strengths

We next examined stability and association strengths in CCA analyses of empirical datasets more generally, through analysis of the published neuroimaging literature using CCA for brain-behavior associations. From 100 CCAs that were reported in 31 publications (see Methods), we extracted the number of observations, number of features, and association strengths. Most studies used <10 samples per feature (Fig. 6a and Supplementary Fig. 2a). Overlaying reported between-set correlations as a function of

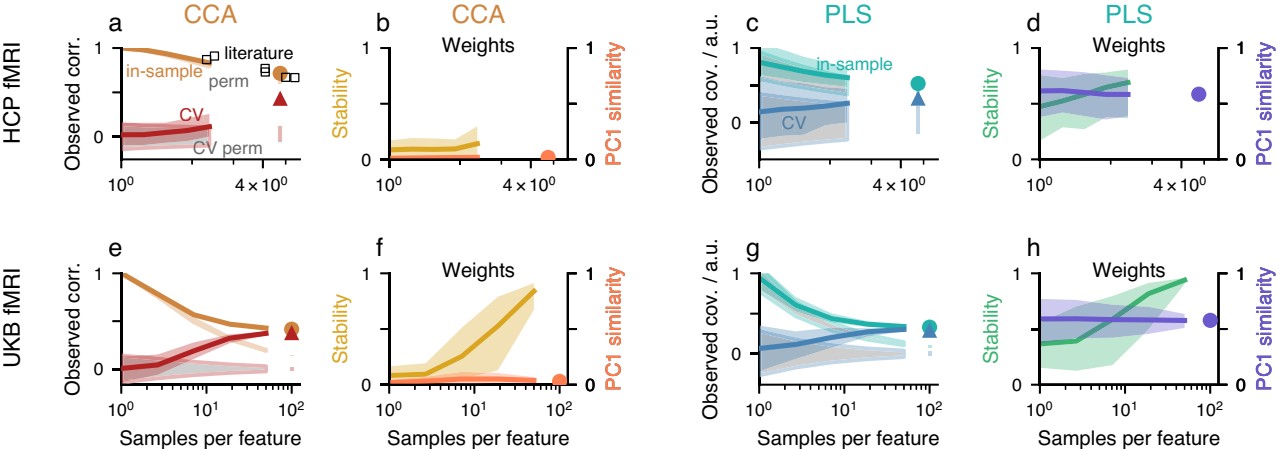

**Fig. 5 | CCA and PLS analysis of empirical population neuroimaging datasets.** For both datasets and for both CCA and PLS a significant mode of association was detected via one-sided permutation testing ($p = 0.001$, $0.003$, $0.001$ and $0.001$ for **a**, **c**, **e**, **g**, respectively). Association strengths monotonically decreased with size of the subsamples (orange in column 1, green in column 3). Association strengths for permuted data ($n_{perm} = 1000$) are shown in gray (with orange and green outlines in columns 1 and 3, respectively). Deviations of the orange and green curves from the gray curves occur for sufficient sample sizes and correspond to significant $p$-values. Note how these curves clearly diverge for UKB but not for HCP data where the number of available subjects is much lower. A circle indicates the estimated value using all available data and the vertical bar in the same color below it denotes the corresponding 95 % confidence interval obtained from permuted data. In (**a**) we also overlaid reported between-set correlations from other studies that used HCP data reduced to 100 principal components. Cross-validated association strengths are shown in red (column 1)

and blue (column 3), cross-validated estimation strengths of permuted datasets in gray with red and blue outlines in columns 1 and 3, respectively. A triangle indicates the cross-validated association strength using all data and the vertical bar in the same color below it denotes the corresponding 95 % confidence interval obtained from permuted data. Cross-validated association strengths were always lower than in-sample estimates and increased with sample size. For UKB (but not yet for HCP) cross-validated association strengths converged to the same value as the in-sample estimate. In the second and fourth columns (panels **b**, **d**, **f** and **h**), weight stabilities (calculated according to Eq. (17)) increased with sample size for UKB and slightly for the PLS analyses of HCP datasets, while they remained low for the CCA analyses of HCP datasets. PC1 weight similarity (calculated according to Eq. (18)) was low for CCA but high for PLS. All analyses were performed with 100 randomly drawn subsamples of varying sizes ($x$-axis). For each subsample size and repetition, we created two non-overlapping sets of subjects and calculated weight stability using these non-overlapping pairs.

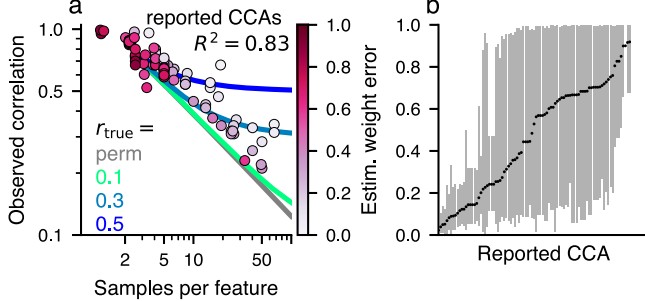

**Fig. 6 | CCAs reported in the population neuroimaging literature might often be unstable. a** Between-set correlations and the number of samples per features are extracted from the literature and overlaid on predictions from the generative model for various true between-set correlations $r_{true}$. The reported between-set correlation can be predicted from the used number of samples per feature alone using linear regression ($R^2 = 0.83$). We also estimated the weight error (encoded in the colorbar) for each reported CCA (details in Supplementary Fig. 16). **b** The distribution of estimated weight errors for each reported CCA is shown along the $y$-axis. For many studies weight errors could be quite large, suggesting that conclusions drawn from interpreting weights might not be robust. See Supplementary Fig. 16 for the procedure for estimation of weight errors.

samples per feature on top of predictions from our generative model shows that most published CCAs are compatible with a range of true between-set correlations, from about 0.5 down to 0 (Fig. 6a). Remarkably, despite the variety of datasets and modalities used in these studies, the reported between-set correlation could be well predicted simply by the number of samples per feature alone ($R^2 = 0.83$) (cf. Supplementary Note 2 and Supplementary Fig. 22 for a corresponding scaling law). We also note that reported CCAs might be biased upwards to some degree due to the fact that researchers might have explored a number of different analyses and reported the one with the highest between-set correlation.

We next asked to what degree weight errors could be estimated from published CCAs. As these are unknown in principle, we estimated them using our generative modeling framework. We did this by (i) generating synthetic datasets of the same size as a given empirical dataset, and sweeping through assumed true between-set correlations between 0 and 1, (ii) selecting those synthetic datasets for which the estimated between-set correlation matches the empirically observed one, and (iii) using the weight errors in these matched synthetic datasets as estimates for weight error in the empirical dataset (Supplementary Fig. 16). This resulted in a distribution of weight errors across the matching synthetic datasets for each published CCA study that we considered. The mean of these distributions is shown in color overlay in Fig. 6a and the range of the distributions is shown in Fig. 6b (see also Supplementary Fig. 2b). These analyses suggest that many published CCA studies likely have unstable feature weights due to an insufficient sample size.

## Calculator for required sample size

How many observations are required for stable CCA/PLS results, given particular dataset properties? One can base this decision on a combination of criteria, by bounding statistical power as well as relative error in association strength, weight error, score error and loading error at the same time. Requiring at least 90 % power and admitting at most 10 % error for other metrics, we determined the corresponding sample sizes in synthetic datasets by interpolating the curves in Fig. 2 (see Supplementary Fig. 17a and Methods). The results are shown in Fig. 7 (see also Supplementary Figs. 18, 19, and 20). For example, when the true between-set correlation is 0.3, several hundreds to thousands of observations are necessary to achieve the indicated power and error bounds (Fig. 7a). The required sample size per feature as a function of the true between-set correlation roughly follows a power-law dependence, with a strong increase in required sample size when the true between-set correlation is low (Fig. 7b). We also evaluated required sample sizes for a commonly used sparse CCA method (Supplementary

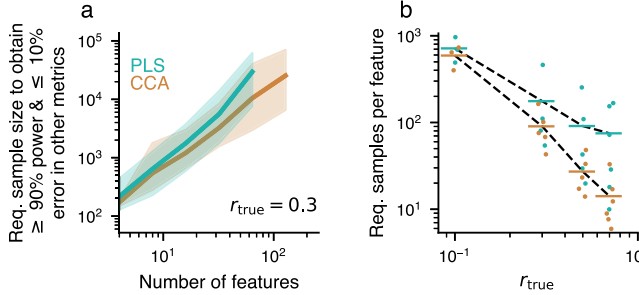

**Fig. 7 | Required sample sizes.** Sample sizes to obtain at least 90 % power and at most 10 % error for the association strength, weight, scores and loadings. Shown estimates are constrained by the within-set variance spectrum (here $a_X + a_Y = -2$, cf. Supplementary Fig. 19 for other values). **a** Assuming a true between-set correlation of $r_{true} = 0.3$ (see Supplementary Fig. 18a–d for other values) 100s to 1000s of observations are required to reach target power and error levels. Shaded areas show 95 % confidence intervals across 25 covariance matrices encoding CCA/PLS solutions with $r_{true} = 0.3$, but varying weight vectors. **b** The required number of observations divided by the total number of features in $X$ and $Y$ scales with $r_{true}$. For $r_{true} = 0.3$ about 50 samples per feature are necessary to reach target power and error levels in CCA, which is much more than typically used (cf. Supplementary Fig. 2a). Every point for a given $r_{true}$ represents a different number of features and is slightly jittered for visibility. Values for a given dimensionality $p_X$ are only shown here if simulations were available for both CCA and PLS. Horizontal lines for each $r_{true}$ represent the mean across the available number of features.

Fig. 21, Supplementary Note 3); however, an in-depth analysis of sparse CCA is beyond the scope of this study.

Finally, we formulated a concise, easy-to-use description of the relationship between model parameters and required sample size. To that end, we fitted a linear model to the logarithm of the required sample size, using logarithms of total number of features and true between-set correlation as predictors (Supplementary Fig. 17). We additionally included a predictor for the decay constant of the within-set variance spectrum, $|a_X + a_Y|$. We found that a simple linear model approach yielded good predictive power for CCA and PLS, which we validated using split-half predictions (Supplementary Fig. 17c, d).

## Discussion
We have used a generative modeling framework to reveal how stability of CCA/PLS solutions depends on dataset properties. Our findings underscore that CCA/PLS stability and statistical significance do not need to coincide (see also[45]). Moreover, for small sample sizes in-sample association strengths severely over-estimate their true value, out-of-sample estimates on the other hand are more conservative. In-sample estimates of association strengths should also not be taken as evidence of predictive validity of a CCA/PLS model. Importantly, estimated weight vectors, which govern the latent feature patterns that underlie an association, do not typically resemble the true weights when the number of observations is low (Fig. 4), which precludes generalizability and interpretability. PLS weights also show a consistent similarity to the first principal component axis (Fig. 4e, f), and therefore PLS weight stability is not sufficient to establish convergence to a true between-set relationship. The same pitfalls appear in state-of-the-art empirical datasets for brain-behavior associations.

CCA/PLS have become popular methods to reveal associations between neuroimaging and behavioral measures[2,3,13,20,40–43,46,47]. The main interest lies in interpreting weights or loadings to understand the profiles of brain imaging and behavioral features carrying the brain-behavior association. We have shown, however, that stability of weights and loadings are contingent on a sufficient sample size which, in turn, depends on the true between-set correlation. How strong are true between-set correlations for typical brain-behavior association studies? While this depends on the dataset at hand and is in principle unknown a priori, Ref. 28 reports average cross-validated (out-of-sample) between-set correlations of 0.17, whereas[30]

argue that higher ($r > 0.2$) out-of-sample between-set correlations are achievable with targeted methods. Our analyses provide insight to this question and highlight the importance of dataset dimensionality. We found in UKB convergence of between-set correlations to ~0.5. As the included behavioral measures comprised a wide assortment of categories, this between-set correlation is likely at upper end of expected ranges. Moreover, we found that most published brain-behavior CCA studies with substantially >10 samples per feature appeared to be compatible only with between-set correlations of ≤0.3, which is at the upper range suggested by recent empirical explorations[28,30].

Assuming a relatively large between-set correlation of 0.3, our generative model still implies that ~50 samples per feature are required for stability of CCA solutions. For designs with hundreds of features, this necessitates many thousands of subjects, in agreement with[28]. Many published brain-behavior CCAs do not meet this criterion. Moreover, in HCP data we saw clear signs that the available sample size was too small to obtain stable solutions—despite that the HCP is one of the largest and highest-quality neuroimaging datasets available to date. On the other hand, in the UKB, where we used 20,000 subjects, CCA and PLS results converged with stability. As UKB-level sample sizes are well beyond what can be feasibly collected in typical neuroimaging studies, these findings support calls for aggregation of datasets that are shared widely[48].

For simplicity and tractability it was necessary to make a number of assumptions in our study. For convenience, we have chosen to represent all data generated by the generative model in each set's principal component coordinate system. This does not affect the validity of the simulations. Moreover, our synthetic data were normally distributed, which is typically not the case in practice. We have assumed a power-law decay model for the within-set variances in each dataset, which we confirmed in a number of empirical datasets (Supplementary Fig. 1), although this might not hold in general. We then assumed the existence of a single cross-modality axis of association, whereas in practice several might be present. In that latter case, theoretical considerations suggest that even larger sample sizes are needed[49]. Additionally, we assumed that the axis of cross-modal association for both the $X$ and $Y$ sets also explains a notable amount of variance within each respective set. While this need not be the case in general, an axis that explains little variance in a set would often not be considered relevant and might not be distinguishable from noise. Importantly, despite these assumptions, empirical brain-behavior datasets yielded similar sample-size dependencies as synthetic datasets.

The numbers of features are important determinants for stability. In our empirical data analysis we have reduced the data to 100 principal components. To be clear, here our goal was to illustrate the behavior of CCA and PLS on a given empirical dataset (i. e. the dataset consisting of the 100 PCs). We do not advocate that taking the first 100 (or any other fixed number, for that matter) of principal components, is an approach that should be taken in practice. Instead, the trade-off between dimensionality-reduction for the purpose of reducing the number of samples required for a stable estimate, and, on the other hand, the effect of a lower canonical correlation as a result of dimensionality reduction requiring, in turn, a higher sample size for stability, needs to be considered. A variety of methods have been proposed to determine an appropriate number of components for PCA and CCA[44,50–53]. Applying one of these methods to HCP data yielded slightly better convergence (Supplementary Fig. 10e–h). Alternatively, prior domain-specific knowledge could be used to preselect features hypothesized to be relevant for the question at hand.

Several related methods have been proposed to potentially circumvent shortcomings of standard CCA/PLS[14]. There exist a number of different PLS variants[54] in addition to the one considered here, which is called PLS-SVD or PLS correlation[10]. They all result in the same first between-set component[54], although note that one variant, PLS regression, is sometimes implemented using an unnormalized first $Y$-weight vector[55]. Higher-order between-set components differ between the PLS variants. Throughout the manuscript we have only considered the first between-set component, which is the one with the highest possible between-set covariance for the given data. Note

that, as required sample sizes for stable estimates depend on the (true) between-set covariance, we expect even higher sample size requirements for all higher-order between-set components than for the first, independent of the PLS variant. Regularized or sparse CCA methods (Supplementary Note 3) apply penalties to weight vectors to mitigate overfitting[56]. We observed that its relative merit might depend on the true weight profile (Supplementary Fig. 21). We also provide an analysis of reduced rank regression in Supplementary Note 4 which suggests it behaves similarly to CCA and PLS (Supplementary Fig. 23). We note that a complete characterization of sparse CCA, reduced rank regression and other methods such as non-linear extensions, was beyond the scope the present study.

In summary, we have presented a parameterized generative modeling framework for CCA and PLS. It allows analysis of the stability of CCA and PLS estimates, prospectively and retrospectively. We end by providing 9 recommendations for using CCA or PLS in practice (Supplementary Table 1).

## Methods

### Experimental design

The goal of this work was to determine requirements for stability of CCA and PLS solutions, both in simulated and empirical data. To do so, we first developed a generative model that allowed us to generate synthetic data with known CCA/PLS solutions. This made it possible to systematically study deviations of estimated from true solutions. Second, we used large state-of-the-art neuroimaging datasets with associated behavioral measurements to confirm the trends that we saw in synthetic data. Specifically, we used data from the Human Connectome Project (HCP) ($n \approx 1000$) and UK Biobank (UKB) ($n = 20,000$). Third, we analyzed published CCA results of brain-behavior relationships to investigate sample-size dependence of CCA results in the literature.

### Human Connectome Project (HCP) data

We used resting-state fMRI (rs-fMRI) from 951 subjects from the HCP 1200-subject data release (03/01/2017)[1]. The HCP source dataset was collected with ethics approval and informed consent from participants[1]. The rs-fMRI data were preprocessed in accordance with the HCP Minimal Preprocessing Pipeline (MPP). Details of the HCP preprocessing can be found elsewhere[57,58]. Following the HCP MPP, BOLD time-series were denoised using ICA-FIX[59,60] and registered across subjects using surface-based multimodal inter-subject registration (MSMAll)[61]. Additionally, global signal, ventricle signal, white matter signal, and subject motion and their first-order temporal derivatives were regressed out[62].

The rs-fMRI time-series of each subject comprised of 2 (69 subjects), 3 (12 subjects), or 4 (870 subjects) sessions. Each rest session was recorded for 15 min with a repetition time (TR) of 0.72 s. We removed the first 100 time points from each of the BOLD sessions to mitigate any baseline offsets or signal intensity variation. We subtracted the mean from each session and then concatenated all rest sessions for each subject into a single time-series. Voxel-wise time series were parcellated to obtain region-wise time series using the "RelatedValidation210" atlas from the S1200 release of the HCP[63]. Functional connectivity was then computed as the Fisher-$z$-transformed Pearson correlation between all pairs of parcels. 3 subjects were excluded (see below), resulting in a total of 948 subjects with 64620 connectivity features each.

Diffusion MRI (dMRI) data and structural connectivity patterns were obtained as described in[64,65]. In brief, 41 major white matter (WM) bundles were reconstructed from preprocessed HCP diffusion MRI data[66] using FSL's XTRACT toolbox[65]. The resultant tracts were vectorised and concatenated, giving a WM voxels by tracts matrix. Further, a structural connectivity matrix was computed using FSL's `probtrackx`[67,68], by seeding cortex/white-gray matter boundary (WGB) vertices and counting visitations to the whole white matter, resulting in a WGB × WM matrix. Connectivity "blueprints" were then obtained by multiplying the latter with the former matrix. This matrix was parcellated (along rows) into 68 regions with

the Desikan-Killany atlas[69] giving a final set of 68 × 41 = 2788 connectivity features for each of the 1020 HCP subjects.

The same list of 158 behavioral and demographic data items as in[3] was used.

We used the following items as confounds: Weight, Height, BPSystolic, BPDiastolic, HbA1C, the third cube of FS_BrainSeg_Vol, the third cube of FS_IntraCanial_Vol, the average of the absolute as well as the relative value of the root mean square of the head motion, squares of all of the above, and an indicator variable for whether an earlier of later software version was used for MRI preprocessing. Head motion and software version were only included in the analysis of fMRI vs behavioral data, not in the analysis of dMRI vs behavioral data. Confounds were inverse-normal-transformed (ignoring missing values) such that each had mean 0. Subsequently, missing values were set to 0. 3% and 5% of confound values were missing in the fMRI vs. behavior, and dMRI vs behavior analysis, respectively. All resulting confounds were $z$-scored once more.

### UK Biobank (UKB) data

We utilized pre-processed resting-state fMRI data[70] from 20,000 subjects, available from the UK Biobank Imaging study[2]. The UKB source dataset was collected with ethics approval and informed consent from participants[70].

In brief, EPI unwarping, distortion and motion correction, intensity normalization and high-pass temporal filtering were applied to each subject's functional data using FSL's Melodic[71], data were registered to standard space (MNI), and structured artifacts are removed using ICA and FSL's FIX[59,60,71]. A set of resting-state networks were identified, common across the cohort using a subset of subjects (≈4000 subjects)[70]. This was achieved by extracting the top 1200 components from a group-PCA[72] and a subsequent spatial ICA with 100 resting-state networks[71,73]. Visual inspection revealed 55 non-artifactual ICA components. Next, these 55 group-ICA networks were dual regressed onto each subject's data to derive representative timeseries for each of the ICA components. Following the regression of the artifactual nodes for all other nodes and the subsequent removal of the artifactual nodes, the timeseries were used to compute partial correlation parcellated connectomes with a dimensionality of 55 × 55. The connectomes were z-score transformed and the upper triangle vectorized to give 1485 functional connectivity features per subject, for each of the 20,000 subjects.

The UK Biobank contains a wide range of subject measures[74], including physical measures (e.g., weight, height), food and drink, cognitive phenotypes, lifestyle, early life factors and sociodemographics. We hand-selected a subset of 389 cognitive, lifestyle and physical measures, as well as early life factors. For categorical items, we replaced negative values with 0, as in[2]. Such negative values encode mostly "Do not know"/"Prefer not to answer". Measures with multiple visits were then averaged across visits, reducing the number of measures to 226. We then performed a check for measures that had missing values in >50% of subjects and also for measures that had identical values in at least 90% of subjects; no measures were removed through these checks. We then performed a redundancy check. Specifically, if the correlation between any two measures was >0.98, one of the two items was randomly chosen and dropped. This procedure further removed 2 measures, resulting in a final set of 224 behavioral measures, available for each of the 20,000 subjects.

We used the following items as confounds: acquisition protocol phase (due to slight changes in acquisition protocols over time), scaling of T1 image to MNI atlas, brain volume normalized for head size (sum of gray matter and white matter), fMRI head motion, fMRI signal-to-noise ratio, age and sex. In addition, similarly to[2], we used the squares of all non-categorical items (i. e. T1 to MNI scaling, brain volume, fMRI head motion, fMRI signal-to-noise ratio and age), as well as age × sex and age$^2$ × sex. Altogether these were 14 confounds. Confounds were inverse-normal-transformed (ignoring missing values), such that each had mean 0. 6% of values were missing and set to 0. All resulting confounds were then $z$-scored across subjects once more.

## Preprocessing of empirical data for CCA and PLS

We prepared data for CCA/PLS following, for the most part, the pipeline in ref. 3.

Deconfounding of a matrix $X$ with a matrix of confounds $C$ was performed by subtracting linear predictions, i.e.

$$X_{\text{deconfounded}} = X - C\beta \tag{1}$$

where

$$\beta = C^+ X = \left(C^{\mathsf{T}} C\right)^{-1} C^{\mathsf{T}} X \tag{2}$$

The confounds used were specific to each dataset and mentioned in the previous section.

Neuroimaging measures were $z$-scored. The resulting data matrix was de-confounded (as described above), decomposed into principle components via a singular value decomposition, and the left singular vectors, multiplied by their respective singular values, were used as data matrix $X$ in the subsequent CCA or PLS analysis. We retained 100 principal components (out of 948, 1020 and 1485 for the HCP-fMRI, HCP-dMRI and UKB analysis, respectively).

The list of used behavioral items were specific to each dataset and mentioned in the previous sections. Given this list, separately for each item, a rank-based inverse normal transformation[75] was applied and the result $z$-scored. For both of these steps subjects with missing values were disregarded. Next, a subjects × subjects covariance matrix across variables was computed, considering for each pair of subjects only those variables that were present for both subjects. Thus, each element of the resulting matrix could, in general, be computed from a different set of subjects such that it is not clear whether the resulting matrix is a proper positive definite covariance matrix. To guarantee a proper covariance matrix for the following, we therefore computed the nearest positive definite matrix of this matrix using the function `cov_nearest` from the Python `statsmodels` package[76]. This procedure has the advantage that subjects can be used without the need to impute missing values. An eigenvalue decomposition of the resulting covariance matrix was performed where the eigenvectors, scaled to have standard deviation 1, are principal component scores. They are then scaled by the square-roots of their respective eigenvalues (so that their variances correspond to the eigenvalues) and used as matrix $Y$ in the subsequent CCA or PLS analysis. We retained 100 (out of 948, 1020 and 20000 for the HCP-fMRI, HCP-dMRI and UKB analysis, respectively) principal components corresponding to the highest eigenvalues.

## Generating synthetic data for CCA and PLS

Note that mathematical derivations and additional explanation of terminology related to CCA and PLS are provided in the Supplementary Information. We analyzed properties of CCA and PLS with simulated datasets from a multivariate generative model. These datasets are drawn from a normal distribution with mean 0 and covariance matrix $\Sigma$ that encodes assumed relationships in the data. To specify $\Sigma$ we need to specify relationships of features within $X$, i. e. the covariance matrix $\Sigma_{XX} \in \mathbb{R}^{p_X \times p_X}$, relationships of features within $Y$, i. e. the covariance matrix $\Sigma_{YY} \in \mathbb{R}^{p_Y \times p_Y}$, and relationships between features in $X$ and $Y$, i.e. the matrix $\Sigma_{XY} \in \mathbb{R}^{p_X \times p_Y}$. Together, these three covariance matrices form the joint covariance matrix (Fig. 1d)"

$$\Sigma = \begin{pmatrix} \Sigma_{XX} & \Sigma_{XY} \\ \Sigma_{XY}^{\mathsf{T}} & \Sigma_{YY} \end{pmatrix} \in \mathbb{R}^{(p_X+p_Y) \times (p_X+p_Y)} \tag{3}$$

for $X$ and $Y$ and this allows us to generate synthetic datasets by sampling from the associated normal distribution $\mathcal{N}(0, \Sigma)$. $p_X$ and $p_Y$ correspond to the number of features in $X$ and $Y$ respectively.

We next describe the covariance matrices $\Sigma_{XX}$ and $\Sigma_{YY}$. Given a data matrix $X$, the features can be re-expressed in a different coordinate system through multiplication by an orthogonal matrix $O$: $\tilde{X} = XO$. No

information is lost in this process, as it can be reversed: $X = \tilde{X}O^{\mathsf{T}}$. Therefore, we are free to make a convenient choice. We select the principal component coordinate system as in this case the covariance matrix becomes diagonal, i. e. $\Sigma_{XX} = \text{diag}(\vec{\sigma}_{XX})$. Analogously, for $Y$ we choose the principal component coordinate system such that $\Sigma_{YY} = \text{diag}(\vec{\sigma}_{YY})$.

For modeling, to obtain a concise description of $\vec{\sigma}_{XX}$ and $\vec{\sigma}_{YY}$ we assume a power-law such that $\sigma_{XX,i} = c_{XX} i^{-a_{XX}}$ and $\sigma_{YY,i} = c_{YY} i^{-a_{YY}}$ with decay constants $a_{XX}$ and $a_{YY}$ (Fig. 1b). Unless a match to a specific dataset is sought, the scaling factors $c_{XX}$ and $c_{YY}$ can be set to 1 as they would only rescale all results without affecting conclusions.

We now turn to the cross-covariance matrix $\Sigma_{XY}$.

For PLS, given a cross-covariance matrix $\Sigma_{XY}$, PLS solutions can be derived via a singular value decomposition as

$$\Sigma_{XY} = USV^{\mathsf{T}} \tag{4}$$

where the singular vectors in the columns of $U$ and $V$ are orthonormal and the matrix $S$ contains the singular values on its diagonal. The columns of $U$ and $V$ are the weight vectors for $X$ and $Y$, respectively, and the singular values give the corresponding between-set association strengths. Thus, the above equation allows us to compute $\Sigma_{XY}$ given orthonormal PLS weight vectors $U$, $V$ and corresponding between-set association strengths $\text{diag}(S)$. This is what we do for PLS. See below for how we select the specific weights that we use. Singular values here in the context of PLS are between-set covariances between $X$ scores and $Y$ scores. We re-express these in terms of a between-set correlation, i.e. the $i$-th singular value, $s_i$, is

$$s_i = r_{\text{true}} \sqrt{\text{var}(X\vec{u}_i)\,\text{var}(Y\vec{v}_i)} \tag{5}$$

where $r_{\text{true}}$ is the assumed true (population) between-set correlation for mode $i$, and $\vec{u}_i$ and $\vec{v}_i$ are the $i$-th columns of $U$ and $V$, respectively.

For CCA, we know that a whitened version of the between-set covariance matrix is related to weight vectors and between-set correlations via the singular value decomposition as

$$\Sigma_{XX}^{-1/2} \Sigma_{XY} \Sigma_{YY}^{-1/2} = USV^{\mathsf{T}}. \tag{6}$$

As above, the singular vectors in the columns of $U$ and $V$ are orthonormal and the matrix $S$ contains the singular values on its diagonal. For CCA, the singular values directly give the between-set correlations, but the singular vectors are not identical to the weight vectors. Instead, CCA weights, $W_X$ and $W_Y$ for $X$ and $Y$, repectively, are given by

$$W_X = \Sigma_{XX}^{-1/2} U \tag{7}$$

$$W_Y = \Sigma_{YY}^{-1/2} V. \tag{8}$$

Thus, given weight vectors as columns of $W_X$ and $W_Y$, as well as population between-set correlations $\text{diag}(S)$, we can calculate the corresponding between-set covariance matrix for CCA as

$$\begin{aligned} \Sigma_{XY} &= \Sigma_{XX}^{1/2} USV^{\mathsf{T}} \Sigma_{YY}^{1/2} \\ &= \Sigma_{XX}^{1/2} \left(\Sigma_{XX}^{1/2} W_X\right) S \left(\Sigma_{YY}^{1/2} W_Y\right)^{\mathsf{T}} \Sigma_{YY}^{1/2} \end{aligned} \tag{9}$$

Importantly, we had assumed that the columns of $U$, as well as $V$, were orthonormal. Thus, the columns of $W_X$ and $W_Y$ must satisfy the following constraints: $U^{\mathsf{T}} U = W_X^{\mathsf{T}} \Sigma_{XX} W_X = I$, as well as $V^{\mathsf{T}} V = W_Y^{\mathsf{T}} \Sigma_{YY} W_Y = I$. One straightforward way to achieve this, is to restrict our consideration to one mode, i. e. we assume that $\Sigma_{XY}$ has rank one (or can be approximated with a rank one matrix) such that $U$, $V$, as well as $W_X$ and $W_Y$ consist of only 1 column. In this case, whatever weight vector we choose, say $\tilde{w}_X$

(analogously for $Y$), we can normalize it as

$$\vec{w}_X = \tilde{\vec{w}}_X / \sqrt{\tilde{\vec{w}}_X^\mathsf{T} \Sigma_{XX} \tilde{\vec{w}}_X} \qquad (10)$$

and $\vec{w}_X$ will satisfy the constraint. See below, for how we select the specific weight vectors we use in our simulations.

As weight vectors, we choose random unit vectors of the desired dimension, as long as they satisfy the following two constraints. For the first constraint, we aim to obtain association modes that explain a relatively large amount of variance in the data, otherwise the resulting scores could be strongly affected by noise. The decision is based on the explained variance of only the first mode and we require that it is $>1/2$ of the average explained variance of a principal component in the dataset, i.e. we require that

$$\vec{w}_X^\mathsf{T} \Sigma_{XX} \vec{w}_X > \frac{1}{2} \frac{\mathrm{tr}\,\Sigma_{XX}}{p_X} \qquad (11)$$

and analogously for $Y$. The weight vectors impact the joint covariance matrix $\Sigma$ (via Eqs. (3), (4) and (9)). For the second constraint, we therefore require that the chosen weights result in a proper, i. e. positive definite, covariance matrix $\Sigma$.

In summary, to generate simulated data for CCA and PLS, we vary the assumed between-set correlation strengths $\vec{\rho}_{XY}$, setting them to select levels, while choosing random weights $W_X$ and $W_Y$. The columns of the weight matrices $W_X$ and $W_Y$ must be mutually orthonormal for PLS, while for CCA they must satisfy $W_X^\mathsf{T} \Sigma_{XX} W_X = W_Y^\mathsf{T} \Sigma_{YY} W_Y = I_m$.

## Performed simulations

Sample-per-feature ratios used in some of the parameter sweep simulations were chosen heuristically, dependent on the given $r_{\mathrm{true}}$, with higher maximal sample-to-feature ratios included the higher $r_{\mathrm{true}}$. For this reason, and as the computional expense also grows with the feature space dimensionality, some simulations did not complete.

For Fig. 2, the left 3 columns of Figs. 3, 7, Supplementary Fig. 3, Supplementary Fig. 6c–f, Supplementary Fig. 7, and Supplementary Fig. 18, we ran simulations for $m = 1$ between-set association mode assuming true between-set correlations of 0.1, 0.3, 0.5, and 0.7, used dimensionalities $p_X = p_Y$ of 2, 4, 8, 16, 32, and 64 as well as 25 different covariance matrices (except for PLS with $p_X = 64$ and $r = 0.7$ where we only had 10 completed simulations; for PLS with $(p_X, r_{\mathrm{true}}) \in \{(0.5, 64), (0.1, 64), (0.1, 32), (0.1, 16)\}$ no simulation completed; for CCA no simulation completed for $(p_X, r_{\mathrm{true}}) \in \{(0.1, 64)\}$). $a_X + a_Y$ was fixed at -2. 100 synthetic datasets were drawn from each instantiated normal distribution. Where not specified otherwise, null distributions were computed with 1000 permutations.

Similar parameters were used for other figures, except for the following deviations.

For the right 3 columns in Fig. 3, $p_X + p_Y$ was fixed at 64, for $p_X$ we used 2, 4, 8, 16, 32 and we constructed 40 different covariance matrices.

For Fig. 4a, b, $p_X$ was 100, $r_{\mathrm{true}} = 0.3$, $a_X = a_Y = -1$, we used 1 covariance matrix for CCA and PLS, and drew 10 collections of observations for each sample size.

For Fig. 4c, d, $p_X$ was 100, $r_{\mathrm{true}} = 0.3$ and we used 100 different covariance matrices.

For Fig. 4e, f, we used 2, 4, 8, 16, 32 and 64 for $p_X$, 0.1, 0.3 and 0.5 for $r_{\mathrm{true}}$, 10 different covariance matrices for CCA and PLS.

For the colored curves in 6a, we used 2, 4, 8, 16, 32, 64 and 128 for $p_X$, true = 0.1, 0.3 and 0.5, generated 10 different covariance matrices and used 10 permutations.

For Fig. 6b, we varied $r_{\mathrm{true}}$ from 0 to 0.99 in steps of 0.01 for each combination of $p_X$ and $p_Y$ for which we have a study in our database of reported CCAs, assumed $a_X = a_Y = 0$, and generated 1 covariance matrix for each $r_{\mathrm{true}}$.

In Supplementary Fig. 4, for $p_X$ we used 4, 8, 16, 32, 64, we generated 10 different covariance matrices for both CCA and PLS and varied $r_{\mathrm{true}}$ from 0 to 0.99 in steps 0.01.

For Supplementary Fig. 5, we used 2, 4, 8, 16 and 32 for $p_X$, and 10 different covariance matrices for both CCA and PLS.

For Supplementary Fig. 8, $p_X$ was 32, $r_{\mathrm{true}} = 0.3$, $a_X = a_Y$ was -1.5, -1.0 and -0.5, we used 25 different covariance matrices, and drew 25 collections of observations for each sample size.

For Supplementary Figs. 17, 19, and 20, we used 25 different covariance matrices (except for PLS with $p_X = 64$ and $r = 0.7$ where we only had 21 completed simulations; for PLS with $(p_X, r_{\mathrm{true}}) \in \{(0.5, 64), (0.1, 64), (0.1, 32), (0.1, 16)\}$ no simulation completed; for CCA no simulation completed for $(p_X, r_{\mathrm{true}}) \in \{(0.1, 64)\}$). For each instantiated joint covariance matrix, $a_X + a_Y$ was chosen uniformly at random between -3 and 0 and $a_X$ was set to a random fraction of the sum, drawn uniformly between 0 and 1.

In Supplementary Fig. 21, we used $p_X = p_Y = 64$, $r_{\mathrm{true}} = 0.3$, $a_X = a_Y = -1.0$. 25 datasets were drawn from the distribution.

## Meta-analysis of prior literature

A PubMed search was conducted on December 23, 2019 using the query (`"Journal Article"[Publication Type]`) `AND` (`fmri[MeSH Terms] AND brain[MeSH Terms]`) `AND` (`"canonical correlation analysis"`) with filters requiring full text availability and studies in humans. In addition, studies known to the authors were considered. CCA results were included in the meta-analysis if they related neuroimaging derived measures (e. g. structural or functional MRI, …) to behavioral or demographic measures (e. g. questionnaires, clinical assessments, …) across subjects, if they reported the number of subjects and the number of features of the data entering the CCA analysis, and if they reported the observed canonical (i.e. between-set) correlation. This resulted in 100 CCA analyses reported in 31 publications.

## The *gemmr* software package

We provide an open-source Python package, called *gemmr*, that implements the generative modeling framework presented in this paper[77]. Among other functionality, it provides estimators for CCA, PLS and sparse CCA; it can generate synthetic datasets for use with CCA and PLS using the algorithm laid out above; it provides convenience functions to perform sweeps of the parameters on which the generative model depends; and it calculates required sample sizes to bound power and other error metrics as described above. For a full description, we refer to the package's documentation.

## Statistics and reproducibility

We use a number of different metrics to evaluate the effects of sampling error on CCA and PLS analyses.

Power measures the capability to detect an existing association. It is calculated when the true between-set correlation is $>0$ as the probability across 100 repeated draws of synthetic datasets from the same normal distribution that the observed between-set association strength (i.e., correlation for CCA, covariance for PLS) of a dataset is statistically significant. Significance is declared if the $p$-value is below $\alpha = 0.05$. The $p$-value is evaluated as the probability that association strengths are greater in the null-distribution of association strengths. The corresponding null-distribution is obtained from performing CCA or PLS on 1000 datasets where the rows of $Y$ were permuted randomly. Power is bounded between 0 and 1 and, unlike for the other metrics (see below), higher values are better.

The relative error of the between-set association strength is calculated as

$$\Delta r = \frac{\hat{r} - r}{r} \qquad (12)$$

where $r$ is the true between-set association strength and $\hat{r}$ is its estimate in a given dataset.

Weight error $\Delta w$ is calculated as 1—absolute value of cosine similarity between observed ($\hat{\vec{w}}$) and true ($\vec{w}$) weights, separately for datasets $X$ and $Y$,

and the greater of the two errors is taken:

$$\Delta w = \max_{s \in \{X,Y\}} \left( 1 - |\text{cossim}(\hat{\vec{w}}_s, \vec{w}_s)| \right) \tag{13}$$

where

$$\text{cossim}(\hat{\vec{w}}_s, \vec{w}_s) = \frac{\hat{\vec{w}}_s \cdot \vec{w}_s}{\| \hat{\vec{w}}_s \| \| \vec{w}_s \|}. \tag{14}$$

The absolute value of the cosine-similarity is used due to the sign ambiguity of CCA and PLS. This error metric is bounded between 0 and 1 and measures the cosine of the angle between the two unit vectors $\hat{\vec{w}}_s$ and $\vec{w}_s$.

Score error $\Delta t$ is calculated as 1—absolute value of Pearson correlation between observed and true scores. The absolute value of the correlation is used due to the sign ambiguity of CCA and PLS. As for weights, the maximum over datasets $X$ and $Y$ is selected:

$$\Delta t = \max_{s \in \{X,Y\}} \left( 1 - \left| \text{corr}_i \left( \hat{t}_{s,i}^{(\text{test})}, t_{si}^{(\text{test})} \right) \right| \right) \tag{15}$$

Each element of the score vector represents an individual observation (e. g. subject). Thus, to be able to compute the correlation between estimated ($\hat{t}$) and true ($\vec{t}$) score vectors, corresponding elements must represent the same individual observation, despite the fact that in each repetition new data matrices are drawn in which the observations have completely different identities. To overcome this problem and to obtain scores, which are comparable across repetitions (denoted $\hat{\vec{t}}^{(\text{test})}$ and $\vec{t}^{(\text{test})}$), each time a set of data matrices is drawn from a given distribution $\mathcal{N}(0, \Sigma)$ and a CCA or PLS model is estimated, the resulting model (i. e. the resulting weight vectors) is also applied to a test-set of data matrices, $X^{(\text{test})}$ and $Y^{(\text{test})}$ (of the same size as $X$ and $Y$) obtained from $\mathcal{N}(0, \Sigma)$ and common across repeated dataset draws. The score error metric $\Delta t$ is bounded between 0 and 1.

Loading error $\Delta\ell$ is calculated as (1 − absolute value of Pearson correlation) between observed (i. e. estimated) and true (i.e., population) loadings. The absolute value of the correlation is used due to the sign ambiguity of CCA and PLS. As for weights, the maximum over datasets $X$ and $Y$ is selected:

$$\Delta\ell = \max_{s \in \{X,Y\}} \left( 1 - \left| \text{corr}_i \left( \hat{\ell}_{s,i}^{(\text{test})}, \ell_{s,i}^{(\text{test})} \right) \right| \right) \tag{16}$$

True loadings are calculated with SI Eq. 3 (replacing the sample covariance matrix in the formula with its population value). Estimated loadings are obtained by correlating data matrices with score vectors (SI Eq. 2). Thus, the same problem as for scores occurs: the elements of estimated and true loadings must represent the same individual observations. Therefore, we calculate loading errors with loadings obtained from test data ($X^{(\text{test})}$ and $Y^{(\text{test})}$) and test scores ($\vec{t}^{(\text{test})}$ and $\hat{\vec{t}}^{(\text{test})}$) that were also used to calculate score errors.

The loading error metric $\Delta\ell$ is bounded between 0 and 1 and reflects the idea that loadings measure the contribution of original data variables to the between-set association mode uncovered by CCA and PLS.

Loadings are calculated by correlating a score vector with a column of a data matrix (this is a correlation across samples). When a coordinate transformation (like a PCA) is applied to the original data matrix, the score vector remains invariant (geometrically, CCA/PLS find a specific optimal direction in feature space, and this direction remains the same independent of in which coordinate system it is expressed in). The data matrix, on the other hand, will be affected by a coordinate transformation. Thus, loadings depend on whether they are calculated with an original data matrix or a PCA-transformed data matrix. The generative model we use generates data in the principal component coordinate system and we do not consider calculating any of the loadings in a different coordinate system.

We quantified weight (or loading) stability, $s$, as the (absolute value of the) cosine-similarity between weights (or loadings) estimated from two independently drawn datasets:

$$s = |\text{cossim}(\vec{v}_1, \vec{v}_2)| \tag{17}$$

where $v_1$ and $v_2$ are the weights (or loadings) estimated from dataset 1 and 2, respectively. When we have more than one pair of datasets, we average weight (or loading) stability (calcualted from one pair) across all pairs of available datasets.

We calculated PC1 similarity, $s^{(\text{PC1})}$ of a vector $\vec{v}$ (e. g. a weight or loading vector) as

$$s^{(\text{PC1})} = \text{cossim}(\vec{v}, \vec{a}) \tag{18}$$

where $\vec{a}$ represents the first principal component axis for the dataset. I. e., for instance, if we calculate PC1 similarity of $X$-weights we use the first principal components of the data matrix $X$ as $\vec{a}$.

To interpret the distribution of cosine similarities between weights and the first principal component axis we compare this distribution to a reference, namely to the distribution of cosine-similarities between a random $n$-dimensional unit vector and an arbitrary other unit vector $\vec{e}$. This distribution $f$ is given by ([https://math.stackexchange.com/questions/2977867/x-coordinate-distribution-on-the-n-sphere](https://math.stackexchange.com/questions/2977867/x-coordinate-distribution-on-the-n-sphere), accessed April 28, 2020):

$$f_n(x) = \frac{\mathrm{d}\, P(X \le x)}{\mathrm{d}\, x} \tag{19}$$

where $P$ denotes the cumulative distribution function for the probability that a random unit-vector has cosine-similarity with $\vec{e}$ (or, equivalently, projection onto $\vec{e}$) $\le x$. For $-1 \le x \le 0$, $P$ can be expressed in terms of the surface area $A_n(h)$ of the $n$-dimensional hyperspherical cap of radius 1 and height $h$ (i. e. $x - h = -1$)

$$P(X \le x) = \frac{A_n(h)}{A_n(2)} \tag{20}$$

where $A_n(2)$ is the complete surface area of the hypersphere and

$$A_n(h) = \frac{1}{2} A_n(2) I\left( h(2 - h); \frac{n-1}{2}, \frac{1}{2} \right) \tag{21}$$

and $I$ is the regularized incomplete beta function. Thus,

$$f_n(x) = \frac{1}{2} \frac{\mathrm{d}\, I}{\mathrm{d}\, x} \left( (x+1)(1-x); \frac{n-1}{2}, \frac{1}{2} \right) \tag{22}$$

$$= \frac{1}{2} \frac{1}{B\left(\frac{n-1}{2}, \frac{1}{2}\right)} (1 - x^2)^{\frac{n-3}{2}} (x^2)^{-1/2} (-2x) \tag{23}$$

$$= \frac{1}{B\left(\frac{n-1}{2}, \frac{1}{2}\right)} (1 - x^2)^{\frac{n-3}{2}} \tag{24}$$

where $B$ is a beta function and

$$f_n(2\tilde{x} - 1) \propto (2 - 2\tilde{x})^{\frac{n-1}{2} - 1} (2\tilde{x})^{\frac{n-1}{2} - 1} \tag{25}$$

$$\propto f_\beta\left( \tilde{x}; \frac{n-1}{2}, \frac{n-1}{2} \right) \tag{26}$$

where $f_\beta$ is the probability density function for the beta distribution. Hence, $2\tilde{X} - 1$ with $\tilde{X} \sim Beta\left(\frac{n-1}{2}, \frac{n-1}{2}\right)$ is a random variable representing the cosine similarity between 2 random vectors (or, equivalently, the projection of a random unit-vector onto another).

## CCA/PLS analysis of empirical data

Permutation-based $p$-values in Fig. 5 and Supplementary Fig. 10 were calculated as the probability that the CCA or PLS association strength of permuted datasets was at least as high as in the original, unpermuted data. Specifically, to obtain the $p$-value, rows of the behavioral data matrix were permuted and each resulting permuted data matrix together with the unpermuted neuroimaging data matrix were subjected to the same analysis as the original, unpermuted data, in order to obtain a null-distribution of between-set associations. 1000 permutations were used.

Due to familial relationships between HCP subjects they are not exchangeable so that not all possible permutations of subjects are appropriate[78]. To account for that, in the analysis of HCP fMRI vs behavioral data, we have calculated the permutation-based $p$-value as well as the confidence interval for the whole-data (but not the subsampled data) analysis using only permutations that respect familial relationships. Allowed permutations were calculated using the functions hpc2blocks and palm_quickperms with default options as described in https://fsl.fmrib.ox.ac.uk/fsl/fslwiki/PALM/ExchangeabilityBlocks(accessed May 18, 2020). No permutation indices were returned for 3 subjects that were therefore excluded from the functional connectivity vs behavior analysis.

Subsampled analyses (Fig. 5) were performed for 5 logarithmically spaced subsample-sizes between 202 and 50% of the total subject number. For each subsample size 100 pairs of non-overlapping data matrices were used.

Cross-validated CCA/PLS analyses were performed in the following way[20]: For each fold, we first estimated the CCA / PLS weights from the given training data. Second we computed CCA/PLS scores for the test data of the fold by applying these estimated weights to the test data, that is we calculate the matrix-matrix product between the estimated weights and the test-data matrix, separately for both $X$ and $Y$. The between-set association in the test set is then given by the correlation or covariance of these resulting scores. We used 5-fold cross-validation.

## Principal component spectrum decay constants

The decay constant of a principal component spectrum (Supplementary Fig. 1a–j) was estimated as the slope of a linear regression (including an intercept term) of log( explained variance of a principal component ) on log( principal component number ). For each dataset in Supplementary Fig. 1a–j we included as many principal components into the linear regression as necessary to explain either 30% or 90% of the variance.

## Determination of required sample size

As all evaluation metrics change approximately monotonically with the samples per feature ratio, we fit splines of degree 3 to interpolate and to determine the number of samples per feature that approximately results in a given target level for the evaluation metric. For power (higher values are better) we target 0.9, for all the other metrics (lower values are better) we target 0.1. Before fitting the splines, all samples per feature are log-transformed and metrics are averaged across repeated datasets from the same covariance matrix. Sometimes the evaluation metrics show non-monotonic behavior and in case the cubic spline results in multiple roots we filter those for which the spline fluctuates strongly in the vicinity of the root (suggesting noise), and select the smallest remaining root $\tilde{n}$ for which the interpolated metric remains within the allowed error margin for all simulated $n > \tilde{n}$, or discard the synthetic dataset if all roots are filtered out. In case a metric falls within the allowed error margin for all simulated $n$ (i. e. even the smallest simulated $n_0$) we pick $n_0$.

We suggest, in particular, a *combined* criterion to determine an appropriate sample size. This is obtained by first calculating samples per feature sizes with the interpolation procedure just described separately for the metrics power, relative error of association strength, weight error, score error and loading error. Then, for each parameter set, the maximum is taken across these five metrics.

## Sample-size calculator for CCA and PLS

Estimating an appropriate sample size via the approach described in the previous section is computationally expensive as multiple potentially large datasets have to be generated and analyzed. To abbreviate this process (see also Supplementary Fig. 17a) we use the approach from the previous section to obtain sample-size estimates for $r_{\text{true}} \in \{0.1, 0.3, 0.5, 0.7, 0.9\}$, $p_X \in \{2, 4, 8, 16, 32, 64, 128\}$, $p_Y = p_X$, and $a_X + a_Y \sim \mathcal{U}(-3, 0)$, $a_X = c(a_X + a_Y)$, and $c \sim \mathcal{U}(0, 1)$, where $\mathcal{U}$ denotes a uniform distribution; that is for $a_X$ and $a_Y$ we first draw a random number to fix their sum, then set $a_X$ to be some fraction ($c \in [0, 1]$) of that sum, and $a_Y$ to be $1 - c$ times that sum. We then fit a linear model to the logarithms of the sample size, with predictors $\log(r_{\text{true}})$, $\log(p_X + p_Y)$, $|a_X + a_Y|$, and including an intercept term.

We tested the predictions of linear model using a split-half approach (Supplementary Fig. 17b–f), i. e. we refitted the model using either only sample-size estimates for $r_{\text{true}} \in \{0.1, 0.3\}$ and half the values for $r_{\text{true}} = 0.5$, or the other half of the data, and tested the resulting refitted model on the remaining data in each case.

## Reporting summary

Further information on research design is available in the Nature Portfolio Reporting Summary linked to this article.

## Data availability

Human Connectome Project and UK Biobank datasets cannot be made publicly available due to data use agreements. Human Connectome Project and UK Biobank are available for researchers to apply for data access. The outcomes of synthetic datasets that were analyzed with CCA or PLS, as well as the source data for graphs are available from[79].

## Code availability

Our open-source Python software package, gemmr, is freely available at https://github.com/murraylab/gemmr[77]. Jupyter notebooks detailing the analyses and generation of figures presented in the manuscript are available as part of the package documentation.

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

## Acknowledgements

This research was supported by NIH grants R01MH112746 (J.D.M.), R01MH108590 (A.A.), R01MH112189 (A.A.), U01MH121766 (A.A.), and P50AA012870 (A.A.); the European Research Council (Consolidator Grant 101000969 to S.N.S. and S.W.); Wellcome Trust grant 217266/Z/19/Z (S.N.S.); a SFARI Pilot Award (J.D.M., A.A.); DFG research fellowship HE 8166/1-1 (M.H.), Medical Research Council PhD Studentship UK MR/N013913/1 (S.W.), NIHR Nottingham Biomedical Research Centre (A.M.). Data were provided by the Human Connectome Project, WU-Minn Consortium (Principal Investigators: David Van Essen and Kamil Ugurbil; 1U54MH091657) funded by the 16 NIH Institutes and Centers that support the NIH Blueprint for Neuroscience Research; and by the McDonnell Center for Systems Neuroscience at Washington University. Data were also provided by the UK Biobank under Project 43822 (PI: S.N.S.). In part, computations were performed using the University of Nottingham's Augusta HPC service and the Precision Imaging Beacon Cluster.

## Author contributions

Conceptualization: M.H., S.W., A.A., S.N.S., J.D.M. Methodology: M.H., J.D.M. Software: M.H. Formal analysis: M.H., S.W., A.M., B.R. Resources: A.A., S.N.S., J.D.M. Data Curation: A.M., J.L.J., A.H. Writing—Original Draft: M.H., J.D.M. Writing—Review and Editing: All authors. Visualization: M.H. Supervision: J.D.M. Project administration: J.D.M. Funding acquisition: A.A., S.N.S., J.D.M.

## Competing interests

The authors declare the following competing interests: M.H. and J.L.J. are currently employed by Manifest Technologies. A.A. and J.D.M. hold equity with Neumora Therapeutics (formerly BlackThorn Therapeutics) and are co-founders of Manifest Technologies. J.D.M. and A.A. are co-inventors on the patent Methods and tools for detecting, diagnosing, predicting, prognosticating, or treating a neurobehavioral phenotype in a subject, U.S. Application No.16/149,903, filed on October 2, 664 2018, U.S. Application for PCT International Application No.18/054, 009 filed on October 2, 2018. A.A., J.D.M. and J.L.J are co-inventors on the patent Systems and Methods for Neuro-Behavioral Relationships in Dimensional Geometric Embedding(N-BRIDGE), PCT International Application No.PCT/US2119/022110, filed March 13, 2019. A.A., J.D.M., M.H. and J.L.L. are co-inventors on the patent Methods of Identifying Subjects for Inclusion and/or Exclusion in a Clinical Trial, Application No.: 63/533,888, filed August 21, 2023. All other authors declare no competing interests.
