## [Peer Review File · Communications Biology]

Reviewers' comments:

Reviewer #1 (Remarks to the Author):

Helmer and colleagues provide extensive empirical data simulation results to carefully compare Canonical Correlation Analysis and Partial Least Squares in data scenarios that are commonly encountered in imaging neuroscience and other areas of biomedicine.

In particular, the authors introduce a generative model that was used to systematically explore parameter dependencies, to assess stability, to calculate required sample sizes in new studies, and to estimate weight stability in previously published studies. Additionally, a calculator of dataset properties is proposed for anticipating the stability of the CCA and PLS estimates. This is a unique contribution to the broader computational biology literature. This urgent assessment will help guide many quantitative investigations in rich datasets with strong correlations, typical for many types of biomedical measurements.

This reviewer has several suggestions that the author may want to consider, before publication of this study:

Major

a) Semantic hygiene: 'sample' is sometimes used in the sense of observation and sometimes in the sense of drawn collection of observations, which could be avoided by consistent usage of the term throughout.

b) Maybe the "loadings" / "score" / "weight" terminology could be changed to something else. The writing in the whole paper appears to be exceptionally clear, but this terminology choice may perhaps be made more crisp, as a service to computational biomedicine community. The reviewer is not claiming that he/she a perfection solution for this point. One possibility is perhaps by systematically using PLS/CCA "vectors" (the latent variable vector), PLS/CCA "variates" (the projections based on the latent variable vector), and simply PLS/CCA "parameter" (a single variable's association as capture by the model, naturally understandable in analogy to ANOVA and other linear regression models), as well as PLS/CCA "inter-set correlation" (used consistently throughout rather than using mere "correlation" often times which from this reviewer's experience is highly confusing for many applied computational scientists).

c) Principal component bases appear to be an important component of the generative model for the empirical data simulation experiments. In the limitations section of the discussion section, please reflect on the role of rotation-invariance as a known weakness of the PCA approach, and whether and how the present profiling analyses are affected by this.

d) When arguing for the "samples per feature" metric, it could help to cite papers that have used explicitly scaling of samples against the number of input variable before:

Varoquaux, G., 2018. Cross-validation failure: small sample sizes lead to large error bars. *Neuroimage* 180, 68-77.

Schulz, M.-A., Yeo, B.T., Vogelstein, J.T., Mourao-Miranda, J., Kather, J.N., Kording, K., Richards, B., Bzdok, D., 2020. Different scaling of linear models and deep learning in UKBiobank brain images versus machine-learning datasets. *Nat Commun* 11, 1-15.

Bzdok, D., Engemann, D.A., Thirion, B., 2020. Inference and Prediction Diverge in Biomedicine. Patterns (Cell Press).

e) As the authors know, PLS is a family of methods instead of a single method. In the limitations section, please reflect on which particular instance of PLS was interrogated in the present study, and whether results can be expected to differ for other popular variants of PLS.

f) One useful extension of the corpus of simulation results could be to relate them to the increasing use of the Allen Human Brain Atlas. Is there perhaps a way to extend the generative model for fake-data simulation to also cover data scenarios that cover thousands of gene expressions with their relation to high-dimensional brain features. It would be great to extend to these increasingly common imaging-transcriptomics settings, if possible.

Minor

a) Neither abstract nor statement appear to mention "fake/synthetic/empirical data simulation" although this probably exactly describes the character of the used methodology.

b) Discussion / Generalizability. Dot missing at end of subtitle.

c) All figures using "r" metric should unpack in the figure caption what metric this symbol is exactly referring to in each case/graph.

d) Semantics: "Projection on CCA/PLS weights..." does not appear sound to this reviewer. In the case of CCA, the resulting canonical variates are the *result* using the already known CCA parameters. Please make the phrasing less ambiguous to future readers of this paper.

Reviewer #2 (Remarks to the Author):

The manuscript COMMSBIO-23-1546 entitled "On stability of Canonical Correlation Analysis and Partial Least Squares with application to brain-behavior associations" presents an analysis of the numerical stability of CCA and PLS methods, based on a data simulation and data resampling approach.

The paper mostly establishes that

- * stability mostly depends on the samples/features ratio
- * both CCA and PLS suffer from limited power and large variance in the estimates
- * PLS shows less variability but more bias, as it tends to converge to the first principal component of the data

Overall, it cautions against the use of these methods in population studies with high-dimensional data.

I find that the paper does a fairly good job at criticizing ill-grounded statistical approaches. It could well have been written 20 years ago, but ---unfortunately, do to the lack of statistical culture of practitioners--- it is still useful.

I think there is still room for improvement.

Major points

1. Regarding the in-sample / out-of-sample distinction, the paper looks like out-of-sample analysis is useful to provide an unbiased (if not conservative) estimate of the multivariate associations between multi-modal data. This point could be made clearer. But the elephant in the room is that CCA/PLS analyses have no predictive validity: they simply inform practitioners about complex multivariate associations. This dramatically limits their interpretability, but the point is that the community does not understand that and tends to interpret such multivariate associations as predictions ('brain activity predicts XXX' is a familiar, yet often ungrounded paper heading). This has to be written clearly somewhere, at least in the discussion section.

Second, I recommend to introduce asymmetric formulations of PLS (aka reduced-rank regression) that are actually multivariate predictive approaches. I believe that they may behave better in terms of convergence, but this is to be checked. If this is the case, that strengthens the paper by opening a clear direction.

Detailed experiments could go to appendix.

2. I think that the proposed data generation approach makes a lot of assumptions that are somewhat arbitrary. For instance, the variance decay model, that the association between hetero-modal variables is driven by mainly one component, and that the associations involve only the high-variance components of the data ("Weight vectors were chosen randomly and constrained such that the X and Y scores explain at least half as much variance as an average principal component in their respective sets."). There is no reason why this should hold on any dataset. For this reason, I think that the simulation-based results are less conclusive than the one based on data resampling, that do not suffer from unfulfilled hypotheses. This should be discussed more seriously.

This being said, I find it intriguing that "the reported canonical correlation could be well predicted simply by the number of samples per feature alone ($R^2 = 0.83$)". This hints at universal scaling laws which would be worth investigating.

3. The figures are overall, but there are several issue:

* Fig.2 should have error bars.

* I really don't see the point of plotting error in log scale. this makes sense when you study the convergence of an algorithm, but it is not the case here.

* Many figures are too cluttered, with too many sub-figures. For instance Fig.3 Reduce this by half, so that one can parse them more easily and put additional figures in suppmat. The 2-dim analysis is completely useless, it sounds toyish and does not bring any useful information.

* Fig 4 sounds useless, as it repeats fig. 3.

* Fig.5 has too many sub-figures with heterogeneous quantities displayed. Each quantity displayed has to be defined by an equation in the main text.

4. The authors rely on preliminary 100-dim reduction by PCA. The only justification for this is that it was done in previous work. This is not acceptable. This modeling choice had to be justified properly. The only reasonable way I see is to run analysis with and without this reduction and compare the outcome.

A related point on p.12 is that the authors mention "optimizing the number of PCs". It is unclear what it involves. I am really worried that this amounts choosing post-hoc the dimensionality that yields the highest association. If this is the case, this is overfit, and should absolutely be removed, since you want by all means not to do that.

5. It is good that the paper ends with practical recommendations.

* However, I disagree about recommendation 1 if the authors do not indicate a clear procedure to *optimize the number of components* that would not overfit the data.

* I also strongly disagree with recommendation 3, average in-sample and cross-validated correlation. First I don't see any valid reason why cross-validated scores would be negatively biased (besides the sample reduction, but we cannot do much about that). Having pessimistic estimates is better than optimistic ones. Validity is more important than sensitivity.

Minor points

Introduction:

* I find it weird to call "neural features" the features that are commonly used in neuroimaging population studies, such as Cortical thickness, GM density etc. It is quite unclear whether how these features actually relate to neural characteristics. I would thus call those "brain imaging features".

* It should be made clear in the introduction that CCA/PLS do not come with confidence intervals on the loadings or associations, so that one has to resort to some bootstrap approach to get such CIs.

* The methods section should indicate clearly how the out-of-sample analysis is performed (references are not enough)

* Discussion: please do not start with "***Our** generative modeling framework"

* Fig S2 should include a legend with p_x values I think. Fig.S2 carries an essential message I think, hence it should be part of the main text.

* I did not understand the following paragraph, please rephrase "Loadings are calculated by correlating scores with data matrices. All synthetic data matrices in this study are based in the principal component coordinate system. In practice, however, this is not generally the case. Nonetheless, as the transformation between principal component and original coordinate system cannot be constrained, here we do not consider this effect."

Contents

1	Reviewer 1	2
1.1	Major	2
1.2	Minor	5
2	Reviewer 2	5
2.1	Major	5
2.2	Minor	21
3	Additional changes	23

We sincerely thank both reviewers for taking the time to evaluate our manuscript and their suggestions for its improvements which we address in the following.

1 Reviewer 1

1.1 Major

Remark 1. *Semantic hygiene: 'sample' is sometimes used in the sense of observation and sometimes in the sense of drawn collection of observations, which could be avoided by consistent usage of the term throughout.*

To avoid any misunderstanding, we have now clarified occurrences of "sample" in the meaning of a single observation using instead the term "observation", and further annotating them as "single" or "individual" where appropriate. On the other hand, we have rephrased occurrences of "sample" in the meaning of a collection of observations by emphasizing them as a "collection" of observations or samples. Other related terms we often use are, "sample size" "in-sample", "out-of-sample", "samples per feature" or "sample covariance matrix". We believe these terms to be unambiguous.

Remark 2. *Maybe the "loadings" / "score" / "weight" terminology could be changed to something else. The writing in the whole paper appears to be exceptionally clear, but this terminology choice may perhaps be made more crisp, as a service to computational biomedicine community. The reviewer is not claiming that he/she a perfection solution for this point. One possibility is perhaps by systematically using PLS/CCA "vectors" (the latent variable vector), PLS/CCA "variates" (the projections based on the latent variable vector), and simply PLS/CCA "parameter" (a single variable's association as capture by the model, naturally understandable in analogy to ANOVA and other linear regression models), as well as PLS/CCA "inter-set correlation" (used consistently throughout rather than using mere "correlation" often times which from this reviewer's experience is highly confusing for many applied computational scientists).*

We completely agree with the reviewer that terminology used in the literature for CCA/PLS properties is not consistent, and that this situation is unfortunate. The terminology we use is one possible one and, as the reviewer notes, we use it consistently throughout the manuscript. Thus, to address this situation, we propose to add the following prominently in the manuscript, at the end of the first Result paragraph, immediately after we introduce the terminology used in the paper:

We note that alternative terminologies exist [7, 13, 16, 17, 19]. CCA/PLS "scores" (as described above) could also be called "variates"; "weights" (as described above) could also be called "vectors"; and "loadings" (as described above) could also be called "parameters". For CCA, the correlation between the score vectors, i.e. the "between-set correlations", are also called "inter-set correlations" or "canonical correlations".

In addition, we went through the manuscript and replaced, wherever applicable, occurrences of "correlation" with "between-set correlation".

Remark 3. *Principal component bases appear to be an important component of the generative model for the empirical data simulation experiments. In the limitations section of the discussion section, please reflect on the role of rotation-invariance as a known weakness of the PCA approach, and whether and how the present profiling analyses are affected by this.*

PCA rotation-invariance does not affect our analyses. We explain why this is the case. For a given (centered) data matrix X , PCA can be performed via the singular value decomposition (SVD):

$$X = USV^T$$

where the columns of V are the weight vectors and US are the scores. When the input vectors (the rows of X) are rotated by R we find for the PCA of XR^T

$$XR^T = US(RV)^T$$

Thus, while the weight vectors change under a rotation R as $V \rightarrow RV$, the PCA scores are rotation invariant. The intuition behind this finding is that when the geometrical object defined by the point cloud of data vectors is rotated as a whole, the PCA weight vectors adapt and take into account the orientation of the point cloud in the ambient space, but the internal structure of the point cloud, captured by the PCA scores, is not affected by a rotation.

With regard to our generative model, let \vec{x} be a data vector. This vector fundamentally represents a given entity and this entity is independent of the choice of coordinate system. In other words, we can choose any coordinate system to represent the data in, and independent of the choice of coordinate system the vector \vec{x} is always fundamentally the same vector, only expressed in a different way. One possible coordinate system is the one in which the population covariance matrix for the data is diagonal (which co-incides with the principal component coordinate system).

For the simulations that we perform, we chose to use this principal component coordinate system because it is convenient (it makes the within-set covariance matrices diagonal). However, this does not affect the validity of the simulations. Indeed, assume we had chosen any arbitrary other coordinate system. Let $\vec{Z} = (\vec{Z}^T, \vec{Y}^T)^T \sim \mathcal{N}(0, \Sigma)$. Moreover, let $\vec{z}_R = h(\vec{z}) = R\vec{z}$ with $R \in SO(p_X + p_Y)$ represent the rotation of the old into the new coordinate system. If f is the density of \vec{Z} and f_R is the density of \vec{Z}_R , then

$$f_R(\vec{z}_R) = f(h^{-1}(\vec{z}_R)) \left| \det \frac{\partial h^{-1}(\vec{z}_R)}{\partial \vec{z}_R} \right|$$

Note that

$$\left| \det \frac{\partial h^{-1}(\vec{z}_R)}{\partial \vec{z}_R} \right| = 1$$

such that

$$f_R(\vec{z}_R) = f(\vec{z})$$

that is, the probability density of \vec{z}_R is identical to the probability density of \vec{z} . In other words, the generative model generates the vector \vec{z} in the principal component coordinate system with the same probability as the vector $\vec{z}_R = R\vec{z}$ in the rotated coordinate system.

We have added the following to the Limitations paragraph in the Discussion section:

For convenience, we have chosen to represent all data generated by the generative model in each set's principal component coordinate system. This does not affect the validity of the simulations.

Remark 4. *When arguing for the “samples per feature” metric, it could help to cite papers that have used explicitly scaling of samples against the number of input variable before:*

-
- Varoquaux, G., 2018. *Cross-validation failure: small sample sizes lead to large error bars.* *Neuroimage* 180, 68-77.
 - Schulz, M.-A., Yeo, B.T., Vogelstein, J.T., Mourao-Miranada, J., Kather, J.N., Kording, K., Richards, B., Bzdok, D., 2020. *Different scaling of linear models and deep learning in UKBiobank brain images versus machine-learning datasets.* *Nat Commun* 11, 1-15.
 - Bzdok, D., Engemann, D.A., Thirion, B., 2020. *Inference and Prediction Diverge in Biomedicine.* *Patterns* (Cell Press).

We have added the following to the Result section:

We used “samples per feature” as an effective sample size parameter to account for the fact that datasets in practice have very different dimensionalities. Others have previously explored the effect of varying samples and features [3, 14, 18]. Fig. R14 shows that power and error metrics for CCA are parameterized well in terms of “samples per feature”, whereas for PLS it is only approximate. Nonetheless, as “samples per feature” is arguably most straightforward to interpret, we presented results in terms of “samples per feature” for both CCA and PLS.

Remark 5. *As the authors know, PLS is a family of methods instead of a single method. In the limitations section, please reflect on which particular instance of PLS was interrogated in the present study, and whether results can be expected to differ for other popular variants of PLS.*

We have added the following to the Discussion section:

There exist a number of different PLS variants [20] in addition to the one considered here, which is called PLS-SVD or PLS correlation [1]. They all result in the same first between-set component [20], although note that one variant, PLS regression, is sometimes implemented using an unnormalized first Y -weight vector [5]. Higher-order between-set components differ between the PLS variants. Throughout the manuscript we have only considered the first between-set component, which is the one with the highest possible between-set covariance for the given data. Note that, as required sample sizes for stable estimates depend on the (true) between-set covariance, we expect even higher sample size requirements for all higher-order between-set components than for the first, independent of the PLS variant.

Remark 6. *One useful extension of the corpus of simulation results could be to relate them to the increasing use of the Allen Human Brain Atlas. Is there perhaps a way to extend the generative model for fake-data simulation to also cover data scenarios that cover thousands of gene expressions with their relation to high-dimensional brain features. It would be great to extend to these increasingly common imaging-transcriptomics settings, if possible.*

The generative model is agnostic about what each simulated variable represents. I.e. as long as the joint distribution can reasonably be assumed to be approximately multivariate-normal, the generative model could in principle be used.

With respect to modeling relationships between brain features and Allen Human Brain Atlas (AHBA) gene expression data specifically we note the following.

AHBA gene expression data is available for 6 subjects, regions of interest covering the whole brain and $\approx 20k$ genes. In principle, we could extract the between-subjects

covariance matrix for a given set of genes for a given set of brain regions, and use this covariance in the generative model. Due to the low number of subjects, the estimation error for this covariance matrix might be fairly high, though. For example, we know that the differential stability (measuring between-subject stability of the gene maps) differs substantially depending on the gene.

Another modeling scenario is conceivable. We could match AHBA gene expression data and brain features along the dimension of brain regions, i.e. we could seek the multivariate relationships between the $n_{\text{ROI}} \times n_{\text{gene}}$ AHBA gene expression data matrix (i.e. the original gene expression data are averaged across subjects) and a $n_{\text{ROI}} \times n_{\text{brain}}$ brain feature data matrix, so that, for each of n_{ROI} regions of interest, we have n_{gene} gene expression values and n_{brain} brain features. This modeling scenario is challenging though, because now the n_{ROI} individual samples are not independent anymore (due to spatial autocorrelation). Moreover, the number of possible samples is limited by the resolution of the used parcellation and cannot even in principle be increased beyond a certain number (dependent on the resolution of the parcellation). We therefore think that this modeling scenario cannot be well represented in the framework used here.

1.2 Minor

Remark 7. *Neither abstract nor statement appear to mention "fake/synthetic/empirical data simulation" although this probably exactly describes the character of the used methodology.*

The abstract now is

[...] To study these issues systematically, we developed a generative modeling framework to simulate synthetic datasets. [...]

Remark 8. *Discussion / Generalizability. Dot missing at end of subtitle.*

We are now no longer using subtitles in the Discussion.

Remark 9. *All figures using "r" metric should unpack in the figure caption what metric this symbol is exactly referring to in each case/graph.*

We have now clarified in figure captions that r_{true} refers to the ground-truth population between-set correlation that was assumed in the generative model used to simulate the corresponding data.

Remark 10. *Semantics: "Projection on CCA/PLS weights..." does not appear sound to this reviewer. In the case of CCA, the resulting canonical variates are the *result* using the already known CCA parameters. Please make the phrasing less ambiguous to future readers of this paper.*

We have rephrased this in Fig. 1 using the terminology suggested by the reviewer in Remark 2. We now say instead:

Projections based on the CCA/PLS weight vectors ...

(As described above, we continue to use the term "weights" or "weight vectors" instead of "latent variable vectors".) See Fig. R1 for the updated figure.

2 Reviewer 2

2.1 Major

Remark 11. *Regarding the in-sample / out-of-sample distinction, the paper looks like out-of-sample analysis is useful to provide an unbiased (if not conservative) estimate of*

Figure R1. Updated figure "Overview of CCA, PLS and the generative model used to investigate their properties".

the multivariate associations between multi-modal data. This point could be made clearer. But the elephant in the room is that CCA/PLS analyses have no predictive validity: they simply inform practitioners about complex multivariate associations. This dramatically limits their interpretability, but the point is that the community does not understand that and tends to interpret such multivariate associations as predictions ('brain activity predicts XXX' is a familiar, yet often ungrounded paper heading). This has to be written clearly somewhere, at least in the discussion section.

We completely agree that canonical correlations should not be mistaken for predictive validity. Regarding this point, and the in-sample / out-of-sample distinction, we have added the following sentence to the first paragraph in the discussion:

Moreover, for small sample sizes in-sample association strengths severely over-estimate their true value, out-of-sample estimates on the other hand are more conservative. In-sample estimates of association strengths should also not be taken as evidence of predictive validity of a CCA/PLS model.

Furthermore, we have rephrased recommendation 3 (Table S1), which now is

In-sample estimates for association strengths are too high.
Out-of-sample estimates are more conservative. In-sample estimates of association strengths should also not be taken as evidence of predictive validity of a CCA/PLS model.

Remark 12. *Second, I recommend to introduce asymmetric formulations of PLS (aka reduced-rank regression) that are actually multivariate predictive approaches. I believe that they may behave better in terms of convergence, but this is to be checked. If this is the case, that strengthens the paper by opening a clear direction. Detailed experiments could go to appendix.*

We thank the reviewer for the idea to also consider reduced rank regression (RRR).

In the following, we first derive theoretically that RRR is expected to behave similarly to CCA and PLS. We then continue to derive a generative model akin to what we have done for CCA and PLS and end with a simulation that illustrates the theoretical results.

RRR linearly predicts a p_Y -dimensional \vec{y} from a p_X -dimensional \vec{x} , using a rank-constrained coefficient matrix C [6, Section 6.3.2]:

$$\vec{y} = C\vec{x} + \text{Error} \quad (1)$$

C is then found using a least-squares approach by minimizing [6, Section 6.3.2]:

$$W = \text{E} [(\vec{y} - C\vec{x})^T \Gamma (\vec{y} - C\vec{x})] \quad (2)$$

where Γ is a positive-definite symmetric parameter matrix. Note that the choice $\Gamma = \Sigma_{YY}^{-1}$ leads to an RRR method that is equivalent (in a specific sense) to CCA, see [6, Section 6.3.2]. The C that minimizes (2) is [6, Section 6.3.2]

$$C = \Gamma^{-\frac{1}{2}} V_q V_q^T \Gamma^{\frac{1}{2}} \Sigma_{YX} \Sigma_{XX}^{-1} \quad (3)$$

where q is the imposed rank of C , Σ_{AB} is the (cross-)covariance matrix between \vec{a} and \vec{b} (both assumed to be de-meant), and the columns of $V_q = (\vec{v}_1, \dots, \vec{v}_q)$ are the eigenvectors corresponding to the q largest eigenvalues (in descending order) of the matrix

$$R = \Gamma^{\frac{1}{2}} \Sigma_{YX} \Sigma_{XX}^{-1} \Sigma_{XY} \Gamma^{\frac{1}{2}}. \quad (4)$$

To analyze sample-size-dependence of RRR in the same way as we have done for CCC and PLS, we need a generative model that lets us simulate \vec{x} and \vec{y} with known RRR-solution. To that end, we note the following observations:

We first rewrite

$$C = \Gamma^{-\frac{1}{2}} \tilde{C} \Sigma_{XX}^{-\frac{1}{2}} \quad (5)$$

with

$$\tilde{C} = V_q V_q^T \Gamma^{\frac{1}{2}} \Sigma_{YX} \Sigma_{XX}^{-\frac{1}{2}} \quad (6)$$

Next, we make the following coordinate substitutions:

- $\vec{x} \rightarrow \vec{x}_W = \Sigma_{XX}^{-\frac{1}{2}} \vec{x}$
- $\vec{y} \rightarrow \vec{y}_\Gamma = \Gamma^{\frac{1}{2}} \vec{y}$

Then,

$$R = \Sigma_{Y_\Gamma X_W} \Sigma_{X_W Y_\Gamma}. \quad (7)$$

Using the singular value decomposition of $\Sigma_{X_W Y_\Gamma}$,

$$\Sigma_{X_W Y_\Gamma} = U S V^T \quad (8)$$

we get

$$R = V S^2 V^T. \quad (9)$$

Thus, the eigenvalues of R are the diagonal entries of S^2 and the corresponding eigenvectors are V , i. e. the right singular vectors of $\Sigma_{X_W Y_\Gamma}$. Also,

$$\tilde{C} = V_q V_q^T \Sigma_{Y_\Gamma X_W} \quad (10)$$

$$= V_q V_q^T V S U^T \quad (11)$$

$$= V_q S_q U_q^T \quad (12)$$

and

$$C = \left(\Gamma^{-\frac{1}{2}} V_q \right) S_q \left(\Sigma_{XX}^{-\frac{1}{2}} U_q \right)^T \quad (13)$$

where S_q is the top left $q \times q$ submatrix of S and U_q represents the first q columns of U .

In summary, to solve RRR, we calculate the singular value decomposition of $\Sigma_{X_W Y_\Gamma} = \Sigma_{XX}^{-\frac{1}{2}} \Sigma_{XY} \Gamma^{\frac{1}{2}} = U^{(\text{RRR})} S^{(\text{RRR})} (V^{(\text{RRR})})^T$, restrict $U^{(\text{RRR})}$, $S^{(\text{RRR})}$ and $V^{(\text{RRR})}$ to the first q components, obtain the RRR "weights" $W_Y^{(\text{RRR})} = \Gamma^{-\frac{1}{2}} V_q^{(\text{RRR})}$ and $W_X^{(\text{RRR})} = \Sigma_{XX}^{-\frac{1}{2}} U_q$, and then the RRR prediction matrix $C_q = W_Y^{(\text{RRR})} S_q^{(\text{RRR})} (W_X^{(\text{RRR})})^T$.

Compare this to

- PLS, where we obtained the weight vectors $W_X^{(\text{PLS})}$ and $W_Y^{(\text{PLS})}$ from the singular value decomposition of $I \Sigma_{XY} I = U^{(\text{PLS})} S^{(\text{PLS})} (V^{(\text{PLS})})^T$ as $W_X^{(\text{PLS})} = I U_q^{(\text{PLS})}$ and $W_Y^{(\text{PLS})} = I V_q^{(\text{PLS})}$
- CCA, where we obtained the weight vectors $W_X^{(\text{CCA})}$ and $W_Y^{(\text{CCA})}$ from the singular value decomposition of $\Sigma_{XX}^{-\frac{1}{2}} \Sigma_{XY} \Sigma_{YY}^{-\frac{1}{2}} = U^{(\text{CCA})} S^{(\text{CCA})} (V^{(\text{CCA})})^T$ as $W_X^{(\text{CCA})} = \Sigma_{XX}^{-\frac{1}{2}} U_q^{(\text{CCA})}$ and $W_Y^{(\text{CCA})} = \Sigma_{YY}^{-\frac{1}{2}} V_q^{(\text{CCA})}$

Thus, $\Gamma = \Sigma_{Y Y}$ leads to an RRR-method with the same sampling properties as CCA. $\Gamma = I$ provides a hybrid between PLS and CCA in which the X variables are internally whitened (as in CCA), but the Y variables are not (as in PLS), and the sampling properties are therefore again not different from CCA and PLS.

To illustrate this with simulations, we next derive a generative model. As we did for CCA and PLS, we will work in the coordinate system in which the within-set covariance matrices are diagonal and assume that the diagonal entries follow a power-law. The between-set covariance matrix, $\Sigma_{X Y}$, is

$$\Sigma_{X Y} = \Sigma_{X X}^{\frac{1}{2}} \Sigma_{X_w Y} \Gamma^{-\frac{1}{2}} \quad (14)$$

$$= \Sigma_{X X}^{\frac{1}{2}} U^{(\text{RRR})} S^{(\text{RRR})} \left(V^{(\text{RRR})} \right)^T \Gamma^{-\frac{1}{2}} \quad (15)$$

$$= \Sigma_{X X}^{\frac{1}{2}} \left(\Sigma_{X X}^{\frac{1}{2}} W_X^{(\text{RRR})} \right) S^{(\text{RRR})} \left(\Gamma^{\frac{1}{2}} W_Y^{(\text{RRR})} \right)^T \Gamma^{-\frac{1}{2}} \quad (16)$$

As $U^{(\text{RRR})}$ and $V^{(\text{RRR})}$ are, respectively, left and right singular vectors, we also need to take into account the following constraints:

$$\left(U^{(\text{RRR})} \right)^T U^{(\text{RRR})} = I \quad \Leftrightarrow \quad \left(W_X^{(\text{RRR})} \right)^T \Sigma_{X X} W_X^{(\text{RRR})} = I \quad (17)$$

$$\left(V^{(\text{RRR})} \right)^T V^{(\text{RRR})} = I \quad \Leftrightarrow \quad \left(W_Y^{(\text{RRR})} \right)^T \Gamma W_Y^{(\text{RRR})} = I \quad (18)$$

We will now make the following further assumptions:

1. $\Sigma_{X Y}$ has rank one, such that $U^{(\text{RRR})}$, $V^{(\text{RRR})}$, $W_X^{(\text{RRR})}$ and $W_Y^{(\text{RRR})}$ all have one column
2. $\Gamma = I$

We can then choose $W_Y^{(\text{RRR})}$ to be any arbitrary unit vector, set $W_X^{(\text{RRR})}$ to be $\vec{w}_X / \sqrt{\vec{w}_X^T \Sigma_{X X} \vec{w}_X}$ for any arbitrary vector \vec{w}_X , and calculate $\Sigma_{X Y}$ from eq. (16).

For CCA, the diagonal entries of $S^{(\text{CCA})}$ were the between-set correlations. For PLS, the diagonal entries of $S^{(\text{PLS})}$ were the between-set covariances, and, to compare PLS with CCA, we have expressed these between-set covariances as a function of the corresponding between-set correlation. This was done by multiplication with the corresponding standard deviations. To compare RRR (with $\Gamma = I$) to CCA and PLS, recall that the X and Y variables turn out to be treated similar as in CCA and PLS, respectively. Thus, by analogy, we will set the diagonal entry of $S^{(\text{RRR})}$ to be the given between-set correlation multiplied by the y -standard deviations (as is done for PLS), but not by the x -standard deviations (as in CCA, the between-set correlation is not multiplied by the standard deviation, to obtain the diagonal of S).

Fig. R2 shows results of a simulation. We have assumed that the between-set correlation is 0.3 and that the dimensionalities of the X and Y feature spaces are both 16. We have also assumed that the within-set covariances decay with a power-law with exponent -1 for both X and Y . We have then generated a random X and a random Y weight vector. For RRR we normalized these vectors according to eqs. (17) and (17), and then constructed the between-set covariance matrix according to (16). Analogously, for PLS, we normalized both of these vectors to have length one and constructed the between-set covariance matrix according to eq. 4 in the Methods. For CCA, on the other hand, we normalized both of these vectors according to the scheme of eq. (17) and constructed the between-set covariance matrix according to eq. 6 in the Methods.

We have then drawn a given number of samples (see x -axis of panels in Fig. R2) from the associated joint normal distribution, estimated S , W_X and W_Y for PLS, CCA

and RRR, and calculated a corresponding error metric. This was repeated 100 times and Fig. R2 shows the mean of the error metrics across these 100 repetitions. Note the similarity between the convergences of PLS, CCA and RRR.

Figure R2. Convergence of PLS, CCA and RRR. Panels show an example simulation with details described in the main text. Note the similarity between PLS, CCA and RRR. **A)** The relative association strength error is calculated as $(\hat{s} - s_{\text{true}})/s_{\text{true}}$, where \hat{s} is the estimated diagonal entry of S and s_{true} is the true value that was assumed when constructing the generative model. **B, C)** The weight error is calculated as $1 - |\text{cossim}(\hat{w}, w_{\text{true}})|$, where cossim denotes cosine-similarity, \hat{w} is the estimated weight vector and w_{true} is the true value that was assumed when constructing the generative model.

We have added the response to this remark to the Supplementary Information.

Remark 13. *I think that the proposed data generation approach makes a lot of assumptions that are somewhat arbitrary. For instance, the variance decay model, that the association between hetero-modal variables is driven by mainly one component, and that the associations involve only the high-variance components of the data (“Weight vectors were chosen randomly and constrained such that the X and Y scores explain at least half as much variance as an average principal component in their respective sets.”). There is no reason why this should hold on any dataset. For this reason, I think that the simulation-based results are less conclusive than the one based on data resampling, that do not suffer from unfulfilled hypotheses. This should be discussed more seriously.*

We have expanded the Discussion paragraph acknowledging limitations of our approach. It now is:

For simplicity and tractability it was necessary to make a number of assumptions in our study. For convenience, we have chosen to represent all data generated by the generative model in each set’s principal component coordinate system. This does not affect the validity of the simulations. Moreover, our synthetic data were normally distributed, which is typically not the case in practice. We have assumed a power-law decay model for the within-set variances in each data set, which we confirmed in a number of empirical data sets (Fig. S1), although this might not hold in general. We then assumed the existence of a single cross-modality axis of association, whereas in practice several might be present. In that latter case, theoretical considerations suggest that even larger sample sizes are needed [10].

Additionally, we assumed that the axis of cross-modal association for both the X and Y sets, also explain a notable amount of variance within each respective set. While this need not be the case in general, an axis that explains little variance in a set would often not be considered relevant and might not be distinguishable from noise. Importantly, despite these assumptions, empirical brain-behavior datasets yielded similar sample-size dependencies as synthetic datasets.

Remark 14. *This being said, I find it intriguing that "the reported canonical correlation could be well predicted simply by the number of samples per feature alone ($R2 = 0.83$)". This hints at universal scaling laws which would be worth investigating.*

In the following we derive, under certain assumptions, a scaling law for the square of the canonical correlation.

When the true population canonical correlations are non-zero, the density of the observed canonical correlations is extremely complicated, following a hypergeometric function of two matrix arguments [2, Section 12.4.2].

Assuming $\Sigma_{XY} = 0$, i.e. that all population canonical correlations are 0, the density for the squares of the observed canonical correlations is given by Anderson [2, Section 13.4]. While this density is still rather complicated in general, some insight can be gained in the special case where one of the feature spaces, say X , has only 1 dimension, i.e. $p_X = 1$.

Let $p = p_X + p_Y$ and let n be the number of available samples. Then, the density d for the squares of the observed canonical correlation, ρ_s , is [2, Section 13.4]

$$d(\rho_s) = \frac{\Gamma[\frac{1}{2}(n-1)]}{\Gamma[\frac{1}{2}(n-p)]\Gamma[\frac{1}{2}(p-1)]} \rho_s^{\frac{1}{2}(p-3)} (1-\rho_s)^{\frac{1}{2}(n-p-2)} \quad (19)$$

From this, we find the expected value of ρ_s as

$$E\rho_s = \int_0^1 \rho_s d(\rho_s) d\rho_s \quad (20)$$

$$= \frac{p-1}{n-1} \quad (21)$$

Empirically, we find this scaling law for the first (highest) observed canonical correlation to hold well for "small" n , even when $p_X > 1$, and even when the true population canonical correlation is > 0 (Fig. R3). $p_X > 1$ leads to a vertical offset in the observed relationship, while the scaling as such appears unchanged.

The scaling law suggests to use $(n-1)/(p-1)$ rather than the quantity "samples per feature", n/p (which we have used in multiple places throughout the manuscript). However, we do not know whether the theoretical scaling with $(n-1)/(p-1)$ would generalize to other quantities like various error metrics, beyond $E r^2$. In fact, we have tried to re-plot Fig. R14 as a function of $(n-1)/(p-1)$ instead of n/p and this did not universally make the curves co-incide more. Moreover, "samples per feature" is conceptually simpler than $(n-1)/(p-1)$. For these reasons, we have chosen to continue to use "samples per feature" throughout the manuscript instead of $(n-1)/(p-1)$.

We have added the response to this remark to the Supplementary Information.

Remark 15. *The figures are overall, but there are several issue:*

1. *Fig.2 should have error bars.*
2. *I really don't see the point of plotting error in log scale. this makes sense when you study the convergence of an algorithm, but it is not the case here.*

Figure R3. Scaling of observed canonical correlation. The expected value for the square of the observed canonical correlation, $E r^2$, largely follows the scaling law (21). Colored lines represent simulation results with different assumed true population canonical correlations. We assumed in the simulations 1 component for the between-set covariance, i.e. one true population covariance $\neq 0$. While the scaling law (black dashed curve) was derived assuming a true between-set correlation of $r_{\text{true}} = 0$ as well as assuming a dimensionality for the X feature space of $p_X = 1$, simulations show that it holds approximately for "small" n for other r_{true} (shown are $r_{\text{true}} = 0.1$ (orange) and 0.3 (green)). $p_X \neq 1$ leads to a vertical offset, while the scaling as such remains constant.

3. Many figures are too cluttered, with too many sub-figures. For instance Fig.3 Reduce this by half, so that one can parse them more easily and put additional figures in suppmat. The 2-dim analysis is completely useless, it sounds toyish and does not bring any useful information.
4. Fig 4 sounds useless, as it repeats fig. 3.
5. Fig.5 has too many sub-figures with heterogeneous quantities displayed. Each quantity displayed has to be defined by an equation in the main text.
1. We have added 95% confidence intervals to Fig. 2. To be able to still discern different curves, we are now only showing $r_{\text{true}}=0.1$ and 0.5 . See Fig. R4.
2. We are now plotting Fig. 2 using linear (instead of log) x and y scales. See Fig. R4.
3. We have reorganized Fig 3, keeping 3 of the previous 6 rows. We have moved the heatmaps in the previous submission's Fig. 3 to the supplementary material. We have dropped the 2-dim analysis. We have also removed the row that showed "Weight PC similarity" as the corresponding point can also be made referring to the first row of this figure. In addition, we have added 95% confidence intervals to the curves in the last row and, to be able to still discern different curves, have opted to remove the curve for permuted data (this does not affect any of the arguments we make). The description of the figure in the Results section has been updated to reflect these changes. See Figs. R5 and R6.

-
4. We have moved Fig. 4 to the supplementary material. We have also re-plotted the first row using linear (instead of log) scales for the x axis, and have added 95% confidence intervals for the loading error (blue). See Fig. R7.
 5. We are now showing only the first and third row of the previous submission's Fig. 5 in the main text, and have moved the other two rows to a supplementary figure. We are now also more explicitly explaining the quantities plotted in the figure. See Figs. R8 and R9.

In addition, to further reduce clutter, and as (per the editorial guidelines) up to 10 display items are allowed in the main text, we have split the figure "CCAs reported in the population neuroimaging literature might often be unstable" (previously Fig. 6) into two parts. See Figs. R10 and R11.

Remark 16. *The authors rely on preliminary 100-dim reduction by PCA. The only justification for this is that it was done in previous work. This is not acceptable. This modeling choice had to be justified properly. The only reasonable way I see is to run analysis with and without this reduction and compare the outcome.*

We acknowledge that, in the submitted manuscript, the justification given for analyzing the dataset restricted to 100 PCs was a reference to previous work. There is, however, a more fundamental reason which we explain in the following.

Some of the data matrices that we start with are extremely high dimensional, having many thousands of features. Given that CCA / PLS stability depend critically on the number of samples, and given that we have only 20,000 subjects in one dataset and only $\approx 1,000$ subjects for others, not reducing the dimensionality would make it impossible for us to analyze how CCA / PLS behave as a function of sample size, while also exploring a meaningful range of sample-per-feature ratios.

The approach we take, thus, is to consider data matrices, X_{100} and Y_{100} for analysis which are comprised of the first 100 principal components of the original (high-dimensional) data matrices, X_{orig} and Y_{orig} . Here, our goal is to learn something about the method, and we emphasize that we make no attempt to interpret the resulting weight or loading vectors. To learn something about the method (CCA or PLS) we use a given empirical dataset, X_{100} and Y_{100} , that is we take the point of view that only X_{100} and Y_{100} are available and given and we disregard the fact that they have been constructed from X_{orig} and Y_{orig} . X_{100} and Y_{100} are perfectly valid data matrices and as such can be analyzed with CCA/PLS, just like all other conceivable data matrices. We could similarly have used any other PCA dimensionality and the trends would be the same. Indeed, this can be seen in Fig. R12E-H (a supplementary figure), where we performed the exact same analyses as in the main text figure but with a smaller number of PCs (we address the question of how we chose the number of PCs in that figure in remark 17; the point we want to emphasize here is simply that we get the same trends with a different number of PCs).

The distinction we emphasize, is that, in a practical application of CCA/PLS, the goal would *not* be to learn something about the method, but to learn something about the data. In that case, we completely agree with the reviewer that any pre-processing to X_{orig} and Y_{orig} that results in an information loss (such as restriction to a given number of principal components) should be considered carefully. For example, dimensionality reduction with PCA has the practical effect of giving CCA less freedom to find the optimal direction in feature space, and as a result the canonical correlation of a PCA-reduced dataset must be smaller than for the original dataset. Thus, in contemplating a dimensionality reduction before CCA, the trade-off to consider is between a higher canonical correlation in a higher-dimensional feature space and a lower canonical correlation in a lower-dimensional feature space, where a greater feature space

Figure R4. Updated figure "Sample-size dependence of CCA and PLS".

Figure R5. Updated figure "Large number of samples required to obtain good weight estimates".

Figure R6. New supplementary figure (formerly part of main text figure "Large number of samples required to obtain good weight estimates").

Figure R7. Updated figure "Stability and PC similarity of weights and loadings".

dimensionality and a lower canonical correlation both command a greater number of samples for stable estimates.

For these reasons, we argue that the presented empirical analyses using 100 principal components is valid in the sense that we have described above. We completely agree with the reviewer that, in practice, where the goal is to learn something about the data, this approach should *not* be taken.

We have clarified the cited paragraph in the Results, where we now say:

We also explored reducing the data to different numbers of PCs than 100. Multiple methods have been proposed to determine an optimal number of PCs (see Discussion). Here, as an example, we used the max-min-detector from [15]. This method suggested 68 brain imaging and 32 behavioral dimensions for HCP [15], which yielded higher cross-validated association strengths and higher stabilities of weights. [...]

Moreover, we have added the following paragraph to the Discussion:

The numbers of features are important determinants for stability. In our empirical data analysis we have reduced the data to 100 principal components. To be clear, here our goal was to illustrate the behavior of CCA and PLS on a given empirical dataset (i.e. the dataset consisting of the 100 PCs). We do not advocate that taking the first 100 (or any other fixed number, for that matter) of principal components, is an approach that should be taken in practice. Instead, the trade-off between dimensionality-reduction for the purpose of reducing the number of samples required for a stable estimate, and, on the other hand, the effect of a lower canonical correlation as a result of dimensionality reduction requiring, in turn, a higher sample size for stability, needs to be considered. A variety of methods have been proposed to determine an appropriate number of components for PCA and CCA [8, 9, 11, 12, 15]. Applying one of these methods to HCP data yielded slightly better convergence (Fig. R12E-H). Alternatively, prior domain-specific knowledge could be used to preselect features hypothesized to be relevant for the question at hand.

Figure R8. Updated figure "CCA and PLS analysis of empirical population neuroimaging datasets". For both datasets and for both CCA and PLS a significant mode of association was detected. Association strengths monotonically decreased with size of the subsamples (orange in column 1, green in column 3). Association strengths for permuted data are shown in grey (with orange and green outlines in columns 1 and 3, respectively). Deviations of the orange and green curves from the grey curves occur for sufficient sample sizes and correspond to significant p -values. Note how these curves clearly diverge for UKB but not for HCP data where the number of available subjects is much lower. Circle indicates the estimated value using all available data and the vertical bar in the same color below it denotes the corresponding 95 % confidence interval obtained from permuted data. In A) we also overlaid reported between-set correlations from other studies that used HCP data reduced to 100 principal components. Cross-validated association strengths are shown in red (column 1) and blue (column 3), cross-validated estimation strengths of permuted datasets in grey with red and blue outlines in columns 1 and 3, respectively. Triangle indicates the cross-validated association strength using all data and the vertical bar in the same color below it denotes the corresponding 95 % confidence interval obtained from permuted data. Cross-validated association strengths were always lower than in-sample estimates and increased with sample size. For UKB (but not yet for HCP) cross-validated association strengths converged to the same value as the in-sample estimate. In the second and fourth columns, weight stabilities (calculated according to eq. 13) increased with sample size for UKB and slightly for the PLS analyses of HCP datasets, while they remained low for the CCA analyses of HCP datasets. PC1 weight similarity (calculated according to eq. 14) was low for CCA but high for PLS. All analyses were performed with repeatedly subsampled data of varying sizes (x -axis). For each subsample size and repetition, we created two non-overlapping sets of subjects and calculated weight stability using these non-overlapping pairs.

Figure R9. New supplementary figure (formerly part of the main text figure "CCA and PLS analysis of empirical population neuroimaging datasets"). Both PC-loadings (calculated as the correlation across observations between CCA/PLS scores and PCs of the data matrices) and original-variable-loadings (calculated as the correlation across observations between CCA/PLS scores and the columns of the data matrices) show a similar pattern as weights (compare Fig. R8), with loadings being slightly more similar to PC1 than weights. Stability and PC1 similarity were calculated according to eqs. 13 and 14, respectively.

Figure R10. Updated figure "CCAs reported in the population neuroimaging literature might often be unstable".

Figure R11. New figure "Required sample sizes".

Remark 17. *A related point on p.12 is that the authors mention "optimizing the number of PCs". It is unclear what it involves. I am really worried that this amounts choosing post-hoc the dimensionality that yields the highest association. If this is the case, this is overfit, and should absolutely be removed, since you want by all means not to do that.*

In principle, we could remove Fig. R12E-H (a supplementary figure) without affecting any of our conclusions. We included Fig. R12E-H to illustrate the potential benefits of PCA before CCA. This is not guaranteed to work in general: if the axis of the between-set association co-incides with the PC that explains the least amount of variance, a dimensionality reduction with PCA would result in the loss of this dimension, and a CCA could no longer discover this dimension. Thus, our intention for Fig. R12E-H is to demonstrate that a pre-CCA PCA, with the number of components chosen by an available method from the literature, *can* be beneficial.

The particular method we have used in Fig. R12E-H is the "max-min-detector" from [15]. This particular method internally exploits, based on certain assumptions, known properties about the sampling distribution of canonical correlations, but other methods have been proposed [8,9,11,12], e.g. based on information theory or cross-validation. We don't want to advocate for one of these methods here, as it is not obvious how these methods behave in comparison. Indeed, we think that such a comparison could be an interesting future direction.

We do acknowledge that we have applied this method to the complete dataset, got the number of components that the method found, restricted the dataset to this number of components, and then created Fig. R12E-H, again using the complete dataset. In practice, when the goal is to learn something about the data instead of the method, we would estimate the number of components in subset of the data, and then estimate the CCA weights on the other part (e.g. in a cross-validation loop).

Remark 18. *It is good that the paper ends with practical recommendations.*

- 1. However, I disagree about recommendation 1 if the authors do not indicate a clear procedure to "optimize the number of components" that would not overfit the data.*
- 2. I also strongly disagree with recommendation 3, average in-sample and cross-validated correlation. First I don't see any valid reason why cross-validated scores would be negatively biased (besides the sample reduction, but we cannot do much about that). Having pessimistic estimates is better than optimistic ones. Validity is more important than sensitivity.*

We have removed the notion of dimensionality reduction from recommendation 1, it now is:

Sample size and the number of features in the dataset are of critical importance for the stability of CCA and PLS.

We have rephrased recommendation 3 to be:

In-sample estimates for association strengths are too high.

Out-of-sample estimates are more conservative. In-sample estimates of association strengths should also not be taken as evidence of predictive validity of CCA/PLS model.

Correspondingly, we have updated the supplementary figure showing cross-validation results. We have removed the curves for the average of in-sample and cross-validation results. Additionally, we have added 95% confidence intervals to the curves in the first two rows. To be able to still discern different curves, we have opted to remove the curves for $r_{\text{true}}=0.3$, and for 20x5-fold CV (which was practically indistinguishable from 5-fold CV). See Fig. R13 for the updated figure.

Figure R12. Additional CCA and PLS analyses of HCP data.

Figure R13. Updated supplementary figure: "Cross-validated estimation of association strength".

2.2 Minor

Remark 19. *Introduction: I find it weird to call "neural features" the features that are commonly used in neuroimaging population studies, such as Cortical thickness, GM density etc. It is quite unclear whether how these features actually relate to neural characteristics. I would thus call those "brain imaging features".*

We have renamed "neural features" to "brain imaging features".

Remark 20. *It should be made clear in the introduction than CCA/PLS do not come with confidence intervals on the loadings or associations, so that one has to resort to some bootstrap approach to get such CIs.*

We have added the following to the second paragraph of the introduction:

Also, while some theoretical results for the sampling properties of CCA are available under normality assumptions [2], one generally needs to resort to resampling approaches to calculate uncertainty estimates like confidence intervals.

Remark 21. *Introduction: The methods section should indicate clearly how the out-of-sample analysis is performed (references are not enough).*

We have added the following to the Methods section:

Cross-validated CCA/PLS analyses were performed in the following way [4]: For each fold, we first estimated the CCA / PLS weights from the given training data. Second we computed CCA/PLS scores for the test data of the fold by applying these estimated weights to the test data, that is we calculate the matrix-matrix product between the estimated weights and the test-data matrix, separately for both X and Y . The between-set association in the test set is then given by the correlation or covariance of these resulting scores. We used 5-fold cross-validation.

Remark 22. *Discussion: please do not start with "***Our** generative modeling framework".*

We have rephrased the first sentence in the discussion to

We have used a generative modeling framework to reveal how stability of CCA/PLS solutions depends on dataset properties.

Remark 23. *Fig S2 should include a legend with p_X values I think. Fig.S2 carries an essential message I think, hence it should be part of the main text.*

We added a legend annotating the p_X values to this figure. See Fig. R14. We have also moved this figure to the main text.

Remark 24. *I did not understand the following paragraph, please rephrase "Loadings are calculated by correlating scores with data matrices. All synthetic data matrices in this study are based in the principal component coordinate system. In practice, however, this is not generally the case. Nonetheless, as the transformation between principal component and original coordinate system cannot be constrained, here we do not consider this effect."*

We have rephrased this as follows:

Figure R14. Updated figure: "Samples per feature is a key effective parameter".

Loadings are calculated by correlating a score vector with a column of a data matrix (this is a correlation across samples). When a coordinate transformation (like a PCA) is applied to the original data matrix, the score vector remains invariant (geometrically, CCA/PLS find a specific optimal direction in feature space, and this direction remains the same independent of in which coordinate system it is expressed in). The data matrix, on the other hand, will be affected by a coordinate transformation. Thus, loadings depend on whether they are calculated with an original data matrix or a PCA-transformed data matrix. The generative model we use generates data in the principal component coordinate system and we do not consider calculating any of the loadings in a different coordinate system.

3 Additional changes

1. Following the editorial guidance that the abstract should be no longer than 150 words, we have slightly shortened it. It now is:

Associations between datasets can be discovered through multivariate methods like Canonical Correlation Analysis (CCA) or Partial Least Squares (PLS). A requisite property for interpretability and generalizability of CCA/PLS associations is stability of their feature patterns. However, stability of CCA/PLS in high-dimensional datasets is questionable, as found in empirical characterizations. To study these issues systematically, we developed a generative modeling framework to simulate synthetic datasets. We found that when sample size is relatively small, but comparable to typical studies, CCA/PLS associations are highly unstable and inaccurate; both in their magnitude and importantly in the feature pattern underlying the association. We confirmed these trends across two neuroimaging modalities and in independent datasets with $n \approx 1000$ and $n = 20000$, and found that only the latter comprised sufficient observations for stable mappings between imaging-derived and behavioral features. We further developed a power calculator to provide sample sizes required for stability and reliability of multivariate analyses.

2. We have made a number of minor edits to improve the manuscript, none of which affect the conclusions.

References

1. H. Abdi and L. J. Williams. Partial Least Squares Methods: Partial Least Squares Correlation and Partial Least Square Regression. In B. Reisfeld and A. N. Mayeno, editors, *Computational Toxicology*, volume 930, pages 549–579. Humana Press, Totowa, NJ, 2013.
2. T. W. Anderson. *An introduction to multivariate statistical analysis*. Wiley series in probability and statistics. Wiley-Interscience, Hoboken, N.J, 3rd ed edition, 2003.
3. D. Bzdok, D. Engemann, and B. Thirion. Inference and Prediction Diverge in Biomedicine. *Patterns*, page 100119, Oct. 2020.

-
4. R. Dinga, L. Schmaal, B. W. J. H. Penninx, M. J. van Tol, D. J. Veltman, L. van Velzen, M. Mennes, N. J. A. van der Wee, and A. F. Marquand. Evaluating the evidence for biotypes of depression: Methodological replication and extension of Drysdale et al. (2017). *NeuroImage: Clinical*, page 101796, Mar. 2019.
 5. A. Höskuldsson. PLS regression methods. *Journal of Chemometrics*, 2(3):211–228, June 1988.
 6. A. Izenman. *Modern Multivariate Statistical Techniques Regression, Classification, and Manifold Learning*. Springer Texts in Statistics. 2008.
 7. V. Kebets, A. J. Holmes, C. Orban, S. Tang, J. Li, N. Sun, R. Kong, R. A. Poldrack, and B. T. T. Yeo. Somatosensory-Motor Dysconnectivity Spans Multiple Transdiagnostic Dimensions of Psychopathology. *Biological Psychiatry*, June 2019.
 8. C. Lameiro and P. J. Schreier. Cross-validation techniques for determining the number of correlated components between two data sets when the number of samples is very small. In *2016 50th Asilomar Conference on Signals, Systems and Computers*, pages 601–605, Nov. 2016. ISSN: null.
 9. Z. Liu, K. J. Whitaker, S. M. Smith, and T. Nichols. Improved Interpretability of Brain-Behavior CCA with Domain-driven Dimension Reduction. *bioRxiv*, page 2021.07.12.451975, July 2021. Publisher: Cold Spring Harbor Laboratory Section: New Results.
 10. A. Loukas. How close are the eigenvectors of the sample and actual covariance matrices? In *Proceedings of the 34th International Conference on Machine Learning - Volume 70*, volume 70, pages 2228–2237. JMLR.org, Aug. 2017.
 11. J. M. Monteiro, A. Rao, J. Shawe-Taylor, and J. Mourão-Miranda. A multiple hold-out framework for Sparse Partial Least Squares. *Journal of Neuroscience Methods*, 271:182–194, Sept. 2016.
 12. P. R. Peres-Neto, D. A. Jackson, and K. M. Somers. How many principal components? stopping rules for determining the number of non-trivial axes revisited. *Computational Statistics & Data Analysis*, 49(4):974–997, June 2005.
 13. R. Rosipal and N. Krämer. Overview and Recent Advances in Partial Least Squares. In C. Saunders, M. Grobelnik, S. Gunn, and J. Shawe-Taylor, editors, *Subspace, Latent Structure and Feature Selection*, Lecture Notes in Computer Science, pages 34–51. Springer Berlin Heidelberg, 2006.
 14. M.-A. Schulz, B. T. T. Yeo, J. T. Vogelstein, J. Mourao-Miranada, J. N. Kather, K. Kording, B. Richards, and D. Bzdok. Different scaling of linear models and deep learning in UKBiobank brain images versus machine-learning datasets. *Nature Communications*, 11(1):1–15, Aug. 2020. Number: 1 Publisher: Nature Publishing Group.
 15. Y. Song, P. J. Schreier, D. Ramírez, and T. Hasija. Canonical correlation analysis of high-dimensional data with very small sample support. *Signal Processing*, 128:449–458, Nov. 2016.
 16. R. M. Thorndike. 9 - Canonical Correlation Analysis. In H. E. A. Tinsley and S. D. Brown, editors, *Handbook of Applied Multivariate Statistics and Mathematical Modeling*, pages 237–263. Academic Press, San Diego, Jan. 2000.

-
17. V. Uurtio, J. M. Monteiro, J. Kandola, J. Shawe-Taylor, D. Fernandez-Reyes, and J. Rousu. A Tutorial on Canonical Correlation Methods. *ACM Computing Surveys (CSUR)*, 50(6):95:1–95:33, Nov. 2017.
 18. G. Varoquaux, P. R. Raamana, D. A. Engemann, A. Hoyos-Idrobo, Y. Schwartz, and B. Thirion. Assessing and tuning brain decoders: Cross-validation, caveats, and guidelines. *NeuroImage*, 145:166–179, Jan. 2017.
 19. H.-T. Wang, J. Smallwood, J. Mourao-Miranda, C. H. Xia, T. D. Satterthwaite, D. S. Bassett, and D. Bzdok. Finding the needle in a high-dimensional haystack: Canonical correlation analysis for neuroscientists. *NeuroImage*, 216:116745, Aug. 2020.
 20. J. A. Wegelin. A survey of Partial Least Squares (PLS) methods, with emphasis on the two-block case. *University of Washington, Department of Statistics, Tech. Rep*, 2000.

REVIEWERS' COMMENTS:

Reviewer #1 (Remarks to the Author):

The authors have done a great job at addressing this reviewer's comments.

Danilo Bzdok MD Phd

Reviewer #2 (Remarks to the Author):

The authors have done a great job addressing my comments. The paper is good for publication; it presents an interesting and useful contribution. Let me simply outline a few minor details:

* Caption of Fig. 2 " betweenw-set"

* Caption of Fig. 5: "Triangle indicates" -> "Triangles indicate" or "The triangle indicates"

* " reducing the number of measures to 226." I guess

* In the **Behavioral and demographic data** paragraph, I miss why the authors need to compute the covariance matrix (with statmodel's cov_nearest function). While this does not look wrong, the motivation is unclear to me.

* Eq (3) you probably want to add parentheses $R^{\{(pX + pY) \times (pX + pY)\}}$

* In the **Sample-size calculator for CCA and PLS** paragraph, I am missing what the authors mean with the $aX = c(aX + aY)$ equation. This defines obviously a relationship between aX and aY , but what is the purpose ?

* Fig. S1 the figure overlaps the caption a little bit.